# A comparison of prospective space-time scan statistics and spatiotemporal event sequence based clustering for COVID-19 surveillance

**Fuyu Xu, Kate Beard** *

School of Computing and Information Science, University of Maine, Orono, ME, United States of America

* kate.beard@maine.edu

## Abstract

The outbreak of the COVID-19 disease was first reported in Wuhan, China, in December 2019. Cases in the United States began appearing in late January. On March 11, the World Health Organization (WHO) declared a pandemic. By mid-March COVID-19 cases were spreading across the US with several hotspots appearing by April. Health officials point to the importance of surveillance of COVID-19 to better inform decision makers at various levels and efficiently manage distribution of human and technical resources to areas of need. The prospective space-time scan statistic has been used to help identify emerging COVID-19 disease clusters, but results from this approach can encounter strategic limitations imposed by constraints of the scanning window. This paper presents a different approach to COVID-19 surveillance based on a spatiotemporal event sequence (STES) similarity. In this STES based approach, adapted for this pandemic context we compute the similarity of evolving daily COVID-19 incidence rates by county and then cluster these sequences to identify counties with similarly trending COVID-19 case loads. We analyze four study periods and compare the sequence similarity-based clusters to prospective space-time scan statistic-based clusters. The sequence similarity-based clusters provide an alternate surveillance perspective by identifying locations that may not be spatially proximate but share a similar disease progression pattern. Results of the two approaches taken together can aid in tracking the progression of the pandemic to aid local or regional public health responses and policy actions taken to control or moderate the disease spread.

## Introduction

The first reported case of Coronavirus disease 2019 (COVID-19) appeared in the US in Washington State in January 2020. Cases then began to appear around the country, creating an outbreak more severe than that experienced in the city of Wuhan, China, where the initial outbreak occurred [1], as well as in many European countries [2, 3]. By mid-March 2020 the outbreak had spread to many states and by late April over one million confirmed cases had been reported in the US.

To anticipate and detect outbreaks, the World Health Organization (WHO), many national and local health departments, academic or other non-profit organizations continuously

**Data Availability Statement:** All data used in the study are available as S1–S5 Tables provided with this submission.

                                                                              

**Funding:** The authors received no specific funding for this work.

**Competing interests:** The authors have declared that no competing interests exist.

collected information about occurrences of COVID-19. Incidence cases were cumulatively added to different online repositories [4–6]. Quick detection of emerging geographical clusters or space-time clusters of COVID-19 can aid public health agencies in prioritizing spatial locations for allocation of different kinds of medical resources including testing kits and applying efficient and publicly acceptable interventions. Versions of space-time scan statistics have been widely used to identify significant clusters of various diseases [7–11] as well as in the current COVID-19 crisis [12, 13]. Space-time scan statistics use circular or elliptical scanning windows of a series of sizes in combination with varying time intervals to systematically scan a study area to detect clusters of disease cases. The Poisson based space-time scan statistic evaluates each scan window for numbers of cases and tests for locations exceeding the number of expected cases under a Poisson distribution.

The prospective Poisson space-time scan statistic has been successfully used for space-time surveillance of different epidemic diseases. As Kulldorff et al. proposed [9, 10], this method focuses on detecting emerging clusters that start at any time during the study period and remain identifiable at the current time (i.e., active or alive), which is the major difference compared to the retrospective space-time scan statistic. Jones et al. used this method to detect twelve "live" or emerging statistically significant (p-value $\leq$ 0.05) clusters of shigellosis in the city of Chicago [14], the results of which helped local health departments to prioritize the assignment and investigation of shigellosis cases. The prospective Poisson space-time scan statistic has also been utilized to identify emerging clusters in other diseases such as thyroid cancer among men in New Mexico (1973–1992) [9], syndromic surveillance [15], measles [16], and dengue fever [17]. More recently, it has been used to detect "active" clusters of COVID-19 confirmed cases in the United States [12, 18].

While the prospective space-time scan statistic is a good option for detecting emerging space-time clusters of infectious diseases, there remain some limitations. The effectiveness of the circular scan window decreases as the shape of emerging clusters becomes more irregular. Detected clusters may contain locations without confirmed cases or with low relative risk due to the artifact of the scanning process [10, 12, 19], although this limitation can be minimized by reporting the individual relative risk for the included locations in each cluster. For the Poisson model, the results depend on accurate data on the population at risk, which may be hard to obtain. Furthermore, the prospective space-time scan statistic as an exploratory method, should be followed with other surveillance measures and more detailed investigation of transmission dynamics and pathogenic mechanics of COVID-19 to better understand detected emerging clusters [12].

While the prospective space-time scan statistic has demonstrated value for COVID-19 surveillance, the objective of this study was to demonstrate a different but complementary view of COVID-19 outbreak patterns. The space time scan statistic detects hotspots but does not inform about locations that may be spatially disparate yet may be exhibiting highly similar patterns in disease case count evolution. To capture this dynamic, we employed an event sequence similarity metric on the sequences of daily COVID incidence rates by county. This event sequence similarity metric was then used to cluster counties exhibiting similarly evolving COVID -19 case histories. The resulting identification of locations exhibiting similar evolutionary patterns in the disease provides another aid for public health responses and understanding of disease dynamics. In the remainder of this paper, we describe this event sequence similarity metric as applied to COVID-19 daily incidence rates and compare it with results of the prospective Poisson space-time scan statistic. We use four time periods to illustrate progression of COVID-19 outbreaks through the lens of prospective space-time scan statistic generated clusters and event sequence similarity clusters. The two approaches provide different but complementary aids to COVID-19 surveillance. One tells us of emerging spatial hotspots,

the other tells us of collections of locations that for some reasons have statistically similar evolving COVID-19 incidence patterns.

## Materials and methods

### Data acquisition and processing

We accessed COVID-19 raw daily global collection data from the GitHub repository (https://github.com/CSSEGISandData/COVID-19) created and maintained by the Johns Hopkins University Center for Systems Science and Engineering (JHU CCSE) [20]. The specific time series dataset for this research contains FIPS codes, state names, geolocations, and confirmed cumulative cases, starting from January 22, 2020 through selected ending dates. JH CCSE continues to semi-automatically or automatically update their site daily (https://raw.githubusercontent.com/CSSEGISandData/COVID-19/master/csse_covid_19_data/).

County level population data for the USA were obtained from the national US Census with estimates for 2019. The ESRI ™ shapefiles of US states and counties used for Geographic Information System (GIS) mapping were downloaded from the TIGER geography portal (US Census Bureau) (https://www.census.gov/cgi-bin/geo/shapefiles/index.php).

We focused the analysis on the 48 contiguous states and Washington D. C.. The dataset was cleaned by filtering out the records without "FIPS" codes and names of counties, and with "FIPS" > 8000 (assigned with "Out of AL", "Out of AK", . . ., "Out of WY"). We combined the cleaned COVID-19 dataset with the U.S. census data at the county level through the "FIPS" codes and double checked the correctness of the spatial information (Latitude and Longitude). Because the COVID-19 dataset only contains cumulative case counts, we obtained the daily confirmed cases by subtracting the previous day's number from the current day's reported cumulative cases. The daily incidence rate for each county was obtained as daily confirmed cases divided by county population and multiplied by 10,000. We chose the data from the first wave of the COVID-19 pandemic in the US in 2020 for this study. The entire duration of the first wave is further divided into four analysis periods considering the incubation time for the disease mostly ranging from 1–14 days with the average of 5 days [21] and the slow case increment at the beginning time in January and February, 2020. The four analysis periods each start from January 22 and cover roughly 2–4 week separations corresponding to an early period 1) March 13, and spiking periods 2) March 31, 3) April 19 and 4) May 20.

### Prospective Poisson space-time scan statistic

We used the prospective Poisson space–time scan statistic as implemented in SaTScan (http://www.satscan.org/) to detect clusters of COVID-19 cases that remained active at the end of each study period. The space–time scan statistic (STSS) is briefly introduced here, and more details can be obtained from [9, 10, 12, 22]. With spatial scan statistics we can identify the locations of clusters of cases. A cluster can be defined as a set of points or regions, at a user defined granularity, with either high or low rates of incidence. For this study, the focus was high rates of COVID-19 incidence. Conceptually the STSS uses a cylinder as the scanning window, where the circular base of the cylinder captures the spatial dimension while the height represents a temporal interval. To identify space-time clusters at the county level, the center of the circular base is co-located with the centroid of each county. As the scan progresses, the radius of the circular base and the height of the cylinder changes from lower bounds to spatial and temporal upper limits. Similar to [12] we set the maximum scanning window base to include up to 10 percent of the total population to avoid the potential of extremely large clusters (ie. covering a quarter of the country) especially as may occur at the beginning stage of the epidemic, and the upper temporal bound to 50% of the entire study period. As each cylinder

moves over the study area, it covers a different set of cases for different time intervals, which can be considered as potential emerging space-time cluster candidates. We set the cluster's duration to a minimum of 2 days and required at least 5 incidents or confirmed cases of COVID-19 as described in [12].

The age structure of a population will influence the incidence of disease, and deaths from COVID-19 are several times higher in older age groups as noted by others [12]. However, we were unable to access age and sex data at this time for cases in this study, so we could not adjust for age and sex. Assuming that COVID-19 incidence follows a Poisson distribution according to the county population, e.g. the assumed population at risk [9], the likelihood ratio test statistic and the relative risk for each scan cylinder was calculated based on the description in [7–9, 12]. The cylinder with the maximum likelihood ratio identifies the location with the most likely elevated risk for COVID-19. We used Standard Monte Carlo simulations (999) in the SaTScan setting to calculate the statistical significance of detected clusters with a p-value equal or less than 0.05 being considered statistically significant. SaTScan computes the relative risk (RR) for each cluster and individual counties. The RR for a county within a cluster can be calculated as in [18]:

$$RR_{cty} = \frac{c/e}{(C-c)(C-e)}$$

Where, c is the total number of cases in a county, C is the total number of observed cases in the conterminous US, and e is the expected number of cases in a county calculated as $e = p_{cty} * \frac{C}{P}$ ($p_{cty}$ is the population in a county, P is the total population). We used ESRI ArcGIS 10.6 (www.esri.com) GIS software to create cartographic representations for these detected emerging clusters at the county level.

## Event sequence similarity-based cluster analysis

Our event sequence similarity approach focuses on the temporal evolution of events occurring at fixed locations. In this study, an event corresponds to the COVID-19 daily incidence rate for a county and a COVID-19 event sequence for a county is the sequence of daily incidence rates covering a specific study period. We compute the similarity of these county level COVID-19 event sequences using a time ordered Jaccard measure [23–25]. Briefly, this measure uses all co-occurrence time points between two event sequences $es_1$ and $es_2$, and calculates the similarity between two events at the co-occurrence timestamp based on their level of measurement. The similarity between two counties' COVID-19 event sequences is calculated as below:

$$sim_{county}(es_1, es_2) = \frac{\sum_{j=1}^{C}(1 - Abs(lev(es_{1j}) - lev(es_{2j})))}{|es_1 \cup es_2|}$$

where,

$sim_{county}(es_1, es_2)$–Similarity between county level event sequences $es_1$ and $es_2$,

$es_{1j}$, $es_{2j}$–the event values for two corresponding co-occurring events in $es_1$ and $es_2$ at timestamp $j$.

$lev(es_{1j})$, $lev(es_{2j})$–the relative event levels of two corresponding co-occurring events in $es_1$ and $es_2$ at timestamp $j$, respectively:

$$lev\left(es_{1j}\right) = \frac{es_{1j}}{es_{1j} + es_{2j}} \text{ and } lev\left(es_{2j}\right) = \frac{es_{2j}}{es_{1j} + es_{2j}}$$

$C$ –the total number of co-occurring timestamps,

$Abs(lev(es_{1j})-lev(es_{2j}))$–absolute value of difference between relative event levels of two corresponding co-occurring events in $es_1$ and $es_2$ at timestamp $j$,

$|es_1 \cup es_2|$–Cardinality of the union of two event sequences $es_1$ $and$ $es_2$.

We then used the computed COVID-19 event sequence similarity measures between counties as the metric for hierarchical clustering [26]. All similarity computations and clustering tasks were implemented in R. The hierarchical clustering was performed using the hclust R function with the linkage method of Ward.D2. The optimal number of clusters was evaluated using the elbow method [27–29]. This method supports selection of the number of clusters at which the total within-cluster sum of square (WSS) no longer improves. In a plot of number of clusters versus WSS, the optimal cluster number is visually associated with the point at which the WSS value flattens.

## Comparison of prospective space time scan and event sequence similarity-based clusters

To support comparison of the two methods we used the counties identified in the prospective Space time scan statistics as having relative risk > 1 as the counties for analysis with the sequence similarity metric. All other counties not included in this set were labeled as OC meaning outside clusters. We include them in Figs 3, 6 and 9 in the graphs of incidences curves for each study period to show their temporal incidence pattern as a baseline.

## Results

### Space-time clusters and sequence similarity-based clusters at county level: Study period 1 (1/22-3/13/2020)

In this early period, COVID-19 was just appearing in the US with the first case reported in Snohomish County Washington on January 19. For this period, the prospective space-time scan statistic identified 11 statistically significant (p-value < 0.05) clusters shown graphically in Fig 1 and summarized in Table 1. These clusters, aside from one in California and two in New York, are generally quite large and counties within them with RR > 1 are few and generally spatially dispersed. Because of the generally large size of these clusters, identifying the spatial specificity of an outbreak is limited.

Based on the elbow evaluation method, 8 event sequence similarity-based clusters were defined for this period (Fig 2). Fig 3 shows the map representation of these clusters along with their temporal profiles. Members of Cluster 3 that include counties in Washington State, California and New York show the earliest onset and the fastest case accumulation. Members of Cluster 5 show an early onset that initially tracks Cluster 3 but then abruptly flattens and then decreases in early March. Members of this cluster include 3 counties in California and one in Minnesota. Cluster 2 members show a delayed occurrence in cases but an extremely fast case accumulation over a few days. The 8 members of this cluster are generally in isolated rural settings in Colorado, Oklahoma, Wyoming, South Dakota, Wisconsin, Louisiana and Indiana. Members of Cluster 6 showed initiation of cases at approximately the same time as Cluster 2 but levelled off quickly at a lower incidence rate. The cluster containing counties in New York suggests initial points of entry and situations conducive to rapid acceleration of cases such as high density or tight knit communities. A pairwise comparison of cluster numbers for the 1[st] study period from these two approaches can be found in S1 Table.

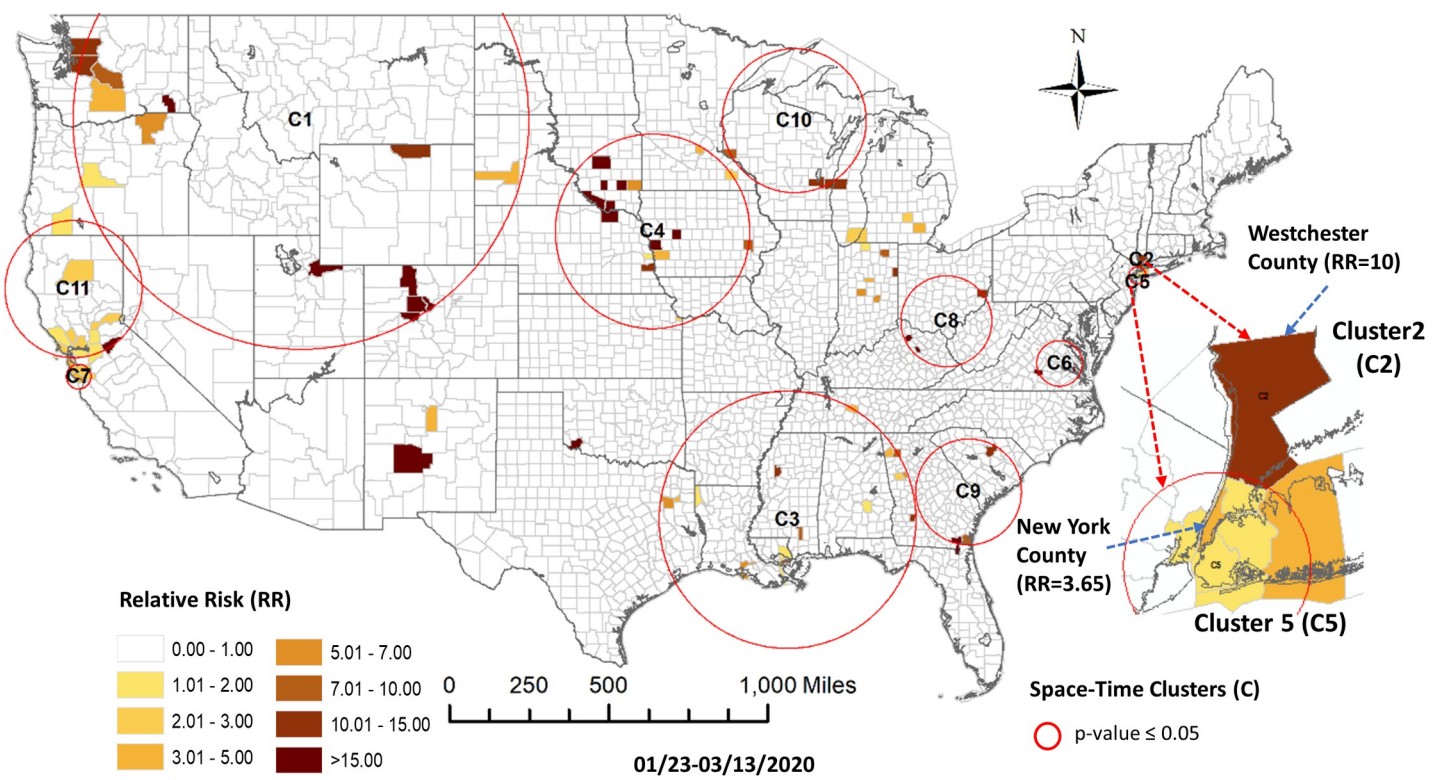

**Fig 1. COVID-19 space-time scan hotspots in the United States at the county level from 1/22/-3/13/2020.**

## Space-time clusters and sequence similarity-based clusters at county level: Study period 2 (1/22-3/31/2020)

Results from the prospective space-time scan statistics analysis for the second study period (through March 31) identified twenty-four space-time clusters of COVID-19 as statistically significant (Fig 4 and Table 2). This period shows a growing emergence of spatial clusters

**Table 1. Attributes of prospective space-time clusters (hotspots) for COVID-19 from 1/23-3/13/2020 at the county level.**

| Cluster | Start Date | End Date | Duration (Days) | Radius (Km) | Observed Cases | Expected Cases | Relative Risk (RR) | p-value | Population at Risk | #County (total) | #County (RR>1) |
|---|---|---|---|---|---|---|---|---|---|---|---|
| 1 | 3/10 | 3/13 | 4 | 806.37 | 389 | 38 | 12.28 | <0.001 | 888,297 | 238 | 14 |
| 2 | 3/7 | 3/13 | 7 | 0.00 | 139 | 15 | 10 | <0.001 | 189,707 | 1 | 1 |
| 3 | 3/10 | 3/13 | 4 | 551.69 | 66 | 18 | 3.83 | <0.001 | 167,447 | 404 | 16 |
| 4 | 3/9 | 3/13 | 5 | 364.08 | 42 | 10 | 4.29 | <0.001 | 87,766 | 262 | 16 |
| 5 | 3/12 | 3/13 | 2 | 32.48 | 102 | 47 | 2.21 | <0.001 | 1,267,395 | 9 | 6 |
| 6 | 3/12 | 3/13 | 2 | 91.08 | 10 | 0 | 29.12 | <0.001 | 7,438 | 35 | 3 |
| 7 | 3/5 | 3/13 | 9 | 49.70 | 93 | 42 | 2.25 | <0.001 | 790,544 | 3 | 3 |
| 8 | 3/9 | 3/13 | 5 | 178.04 | 9 | 0 | 26.67 | <0.001 | 2,607 | 94 | 3 |
| 9 | 3/10 | 3/13 | 4 | 224.18 | 12 | 1 | 14.16 | <0.001 | 15,926 | 104 | 3 |
| 10 | 3/10 | 3/13 | 4 | 253.24 | 12 | 1 | 10.51 | <0.001 | 8,832 | 64 | 3 |
| 11 | 3/7 | 3/13 | 7 | 264.34 | 88 | 47 | 1.91 | <0.001 | 824,139 | 36 | 12 |

Note: Space-time clusters were identified using the spatial scan statistic with a Poisson model.

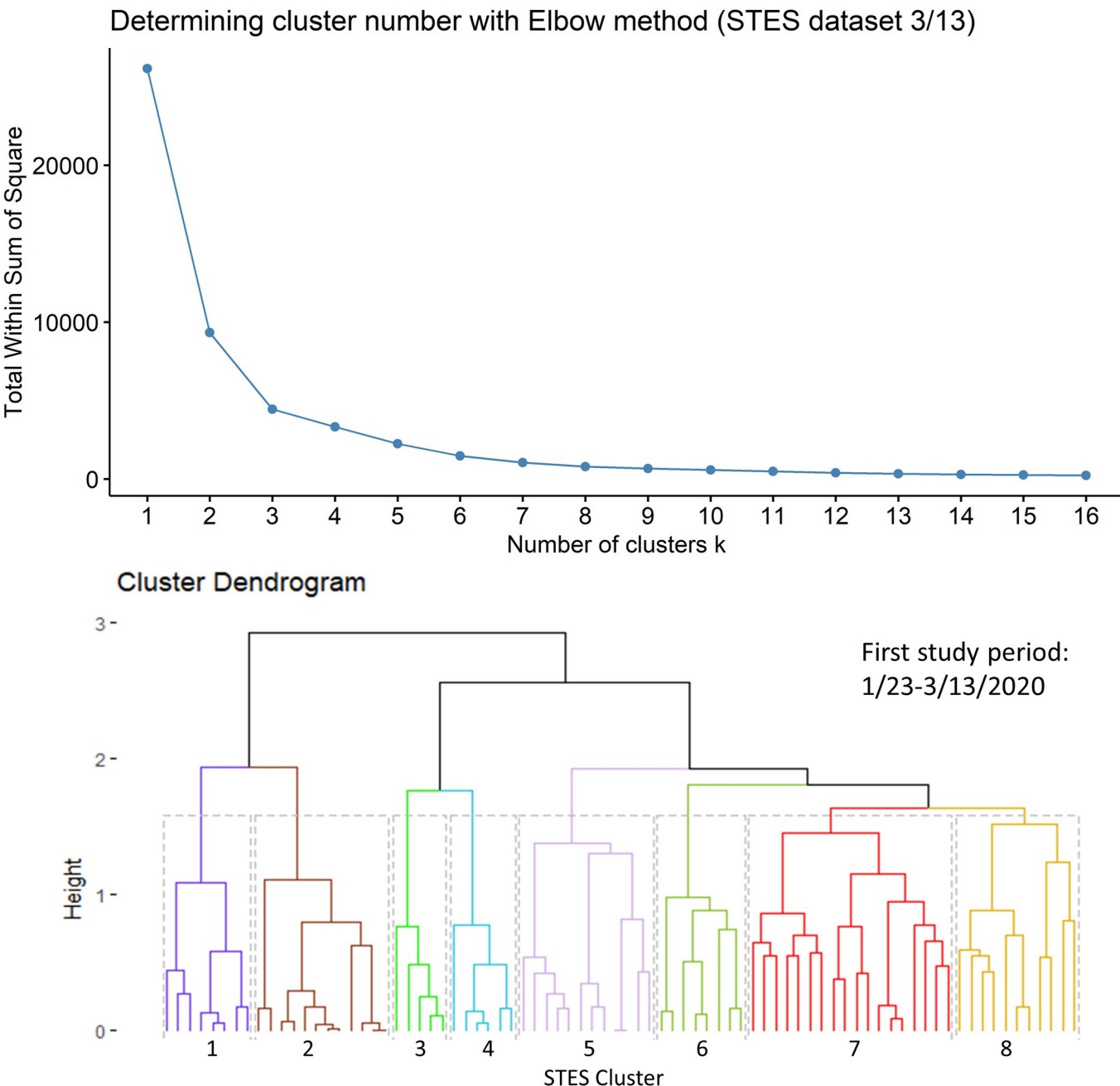

**Fig 2. Elbow method evaluation and hierarchical clustering results for the 1st period.** Notice that the numberings and colors of STES clusters match with those of corresponding clusters on the map and the temporal trend graph in Fig 3.

across the US, but generally more consolidated clusters as the number of cases grow. The space-time clusters are smaller than in the first period and several detected clusters contain a single county (cluster radius = 0). This period shows a shift toward more clusters appearing in the interior US relative to the coasts.

For this second study period the sequence similarity clustering resulted in 8 clusters based on the elbow method evaluation (Fig 5). Fig 6 shows the map of these clusters and their temporal signatures. For this period, only three clusters deviate from the outside cluster (OC) set

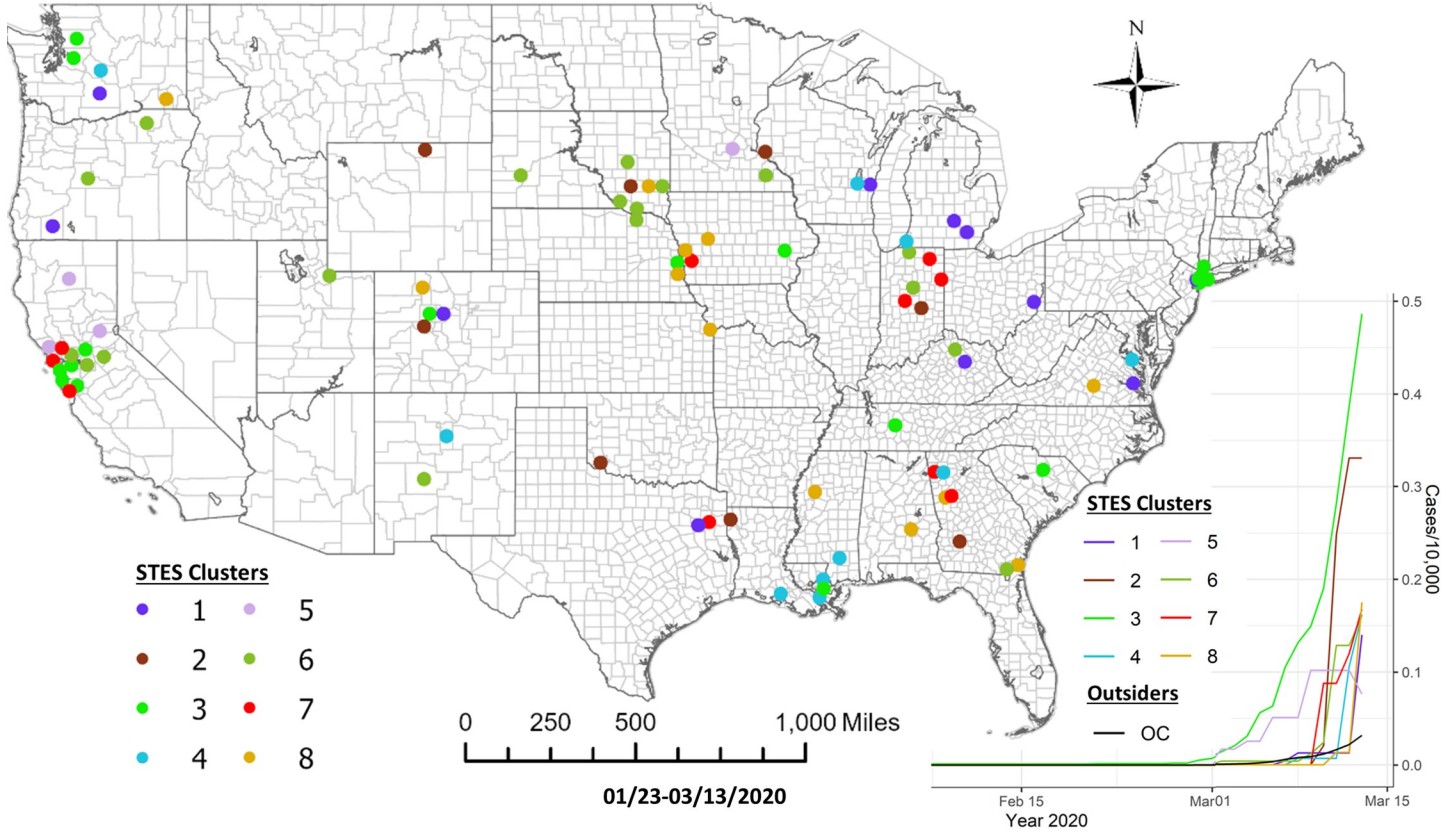

**Fig 3. Sequence similarity-based COVID-19 clusters along with average temporal trends at the county level through 3/13/2020.** This map includes the counties with higher relative risk (RR>1) contained in all the clusters detected by scan statistics in Fig 1. The average temporal trends of cumulative cases for STES clusters 1–8 on the map appear at the bottom right. Notice that the colors of STES clusters match with correspondingly colored dots on the map and with the colors of the STES cluster curves on the graph. OC includes all counties not included in the clusters.

pattern. Cluster 7 shows the most rapid increase in cases. Members of this cluster include Miami, San Jose, Los Angeles area counties, Chicago, Detroit, New Orleans and New York metropolitan counties. Members of Cluster 8 show a slower and less rapid increase in cases. Some of these members appear in a group across New Jersey and Pennsylvania, around Baltimore, Denver and Seattle. Cluster 4 follows a similar trajectory with some concentrations around New Orleans, Columbus Georgia, and Indianapolis. Members of this cluster also appear in more isolated rural settings in Arizona, Oklahoma and South Dakota. A pairwise comparison of cluster numbers for the 2nd study period from these two approaches can be found in S2 Table.

## Space-time clusters and sequence similarity-based clusters at county level: Study period 3 (1/22-4/19/2020)

For the third study period, the prospective space-time cluster statistic detected 47 statistically significant clusters (p≤0.05) as shown in Fig 7. Associated cluster characteristics are shown in Table 3. In this period more clusters are emerging in the southern US, with additional new pockets in Montana and a cluster covering Nebraska and South Dakota. Metropolitan New York remains an active cluster and a more condensed Mid-Atlantic coast cluster has emerged. We see additional consolidation in the size of clusters with 25 appearing as a single county.

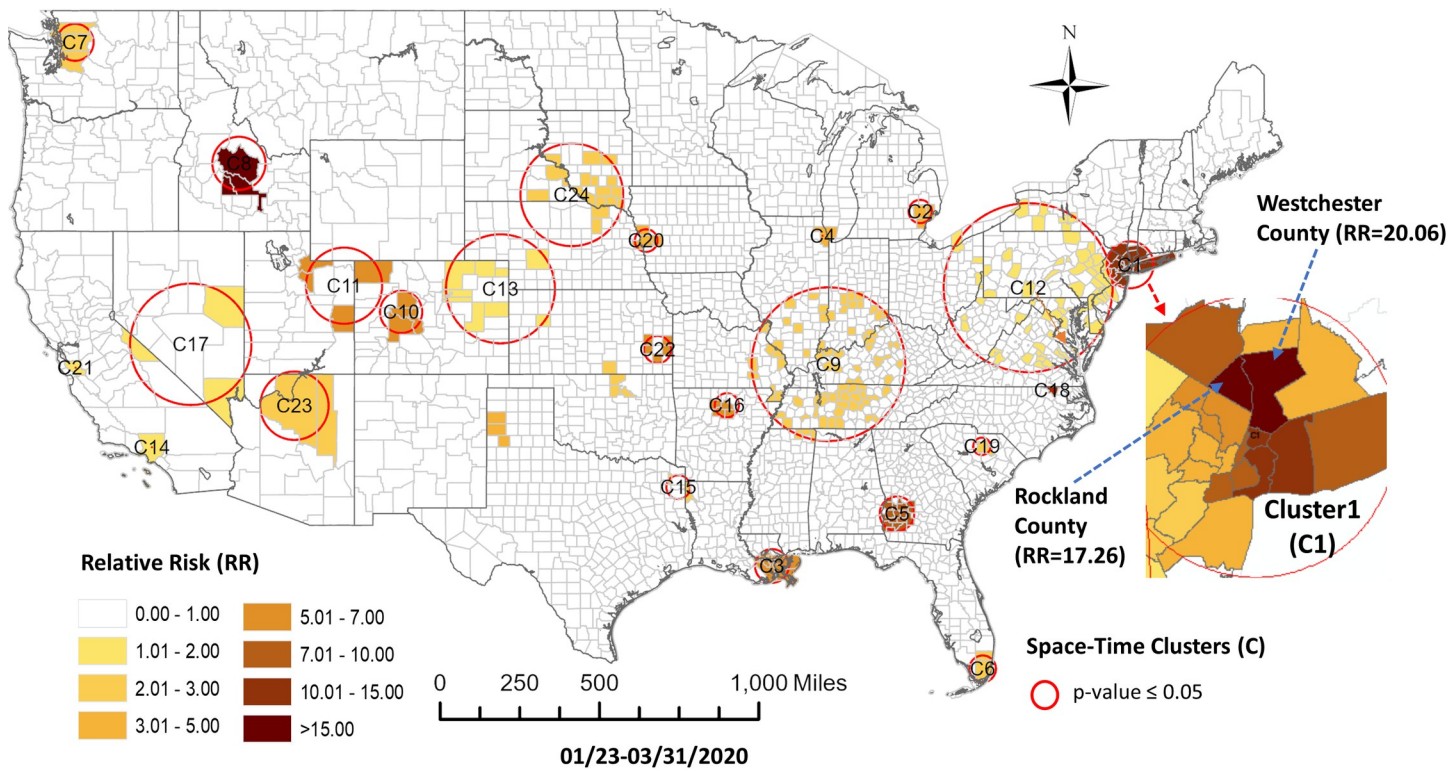

**Fig 4. COVID-19 space-time scan statistic detected hotspots in the United States at county level through 3/31/2020.**

For the third study period, ten sequence similarity-based clusters were selected using the elbow method (Fig 8). Fig 9 shows the map of these clusters and their temporal profiles. Cluster 8 shows a distinct early and more rapid accumulation of cases. Many members of this cluster were members of Cluster 7 in the previous study period. These members include Chicago, Detroit metropolitan area, Miami, Philadelphia, and metropolitan New York counties. Some significant missing members in Cluster 8 from the previous period Cluster 7 are San Jose, Los Angeles and Las Vegas. Cluster 9 shows a group with the next most rapidly developing number of cases. Within this group, some members appear concentrated around metropolitan New York, Philadelphia, Baltimore and Washington DC, and Denver. Cluster 10, as the third most rapidly merging cluster for this period, has members in a halo like pattern around metropolitan New York, Philadelphia and New Orleans. Other members, however, appear in more isolated rural settings in New Mexico, Utah, and Washington State. This group includes the Hopi, Zuni, Navajo and Yakima national reservations. Two other clusters to note in this group are Cluster 7 and Cluster 2 which show later initiation times in terms of case accumulation but appear to be accelerating at the end of the study period. Many of these members show a concentration in southern Indiana and western Kentucky respectively, with another grouping of Cluster 7 members appearing in southwestern Georgia on the border with Alabama. A complete pairwise comparison of cluster numbers for the 3rd study period from these two approaches can be found in S3 Table.

## Space-time clusters and sequence similarity-based clusters at county level: Study period 4 (1/22-5/20/2020)

For the fourth study period ending on May 20, 2020 the prospective space-time scan statistic identified 87 statistically significant clusters. Table 4 provides the characteristics of these 87

**Table 2. Attributes of prospective space-time clusters (hotspots) for COVID-19 from 1/23-3/31/2020 at the county level.**

| Cluster | Start Date | End Date | Duration (Days) | Radius (Km) | Observed Cases | Expected Cases | Relative Risk (RR) | p-value | Population at Risk | #County (total) | #County (RR>1) |
|---|---|---|---|---|---|---|---|---|---|---|---|
| 1 | 3/22 | 3/31 | 13 | 89.28 | 82,928 | 10,049 | 14.35 | <0.001 | 6,395,723 | 22 | 22 |
| 2 | 3/22 | 3/31 | 10 | 43.08 | 5,887 | 1,526 | 3.95 | <0.001 | 1,074,213 | 3 | 3 |
| 3 | 3/20 | 3/31 | 12 | 73.70 | 3,152 | 487 | 6.57 | <0.001 | 292,363 | 8 | 8 |
| 4 | 3/27 | 3/31 | 5 | 0.00 | 3,078 | 1,012 | 3.08 | <0.001 | 2,201,911 | 1 | 1 |
| 5 | 3/24 | 3/31 | 8 | 73.96 | 680 | 68 | 9.97 | <0.001 | 39,490 | 20 | 18 |
| 6 | 3/26 | 3/31 | 6 | 60.42 | 2,587 | 1,102 | 2.37 | <0.001 | 1,370,768 | 2 | 2 |
| 7 | 3/24 | 3/31 | 8 | 62.27 | 2,041 | 846 | 2.43 | <0.001 | 1,345,457 | 4 | 4 |
| 8 | 3/19 | 3/31 | 13 | 95.88 | 190 | 11 | 17.17 | <0.001 | 5,083 | 4 | 3 |
| 9 | 3/30 | 3/31 | 2 | 307.75 | 1,528 | 729 | 2.11 | <0.001 | 1,822,585 | 262 | 82 |
| 10 | 3/16 | 3/31 | 16 | 82.42 | 313 | 54 | 5.78 | <0.001 | 28,677 | 5 | 5 |
| 11 | 3/20 | 3/31 | 12 | 146.72 | 214 | 38 | 5.6 | <0.001 | 20,460 | 9 | 4 |
| 12 | 3/29 | 3/31 | 3 | 325.81 | 4,574 | 3,543 | 1.3 | <0.001 | 6,684,959 | 257 | 75 |
| 13 | 3/27 | 3/31 | 5 | 210.38 | 787 | 448 | 1.76 | <0.001 | 647,610 | 43 | 10 |
| 14 | 3/30 | 3/31 | 2 | 0.00 | 1,190 | 789 | 1.51 | <0.001 | 3,855,599 | 1 | 1 |
| 15 | 3/25 | 3/31 | 7 | 50.46 | 206 | 72 | 2.88 | <0.001 | 57,714 | 5 | 2 |
| 16 | 3/23 | 3/31 | 9 | 49.14 | 84 | 14 | 5.86 | <0.001 | 5,999 | 5 | 4 |
| 17 | 3/30 | 3/31 | 2 | 240.79 | 344 | 179 | 1.92 | <0.001 | 528,991 | 11 | 3 |
| 18 | 3/29 | 3/31 | 3 | 0.00 | 27 | 2 | 11.75 | <0.001 | 1,412 | 1 | 1 |
| 19 | 3/14 | 3/31 | 18 | 36.13 | 105 | 44 | 2.4 | <0.001 | 20,986 | 2 | 2 |
| 20 | 3/22 | 3/31 | 10 | 42.64 | 35 | 8 | 4.27 | <0.001 | 3,227 | 4 | 4 |
| 21 | 3/30 | 3/31 | 2 | 0.00 | 244 | 152 | 1.61 | <0.001 | 991,866 | 1 | 1 |
| 22 | 3/24 | 3/31 | 8 | 54.38 | 22 | 4 | 5.76 | <0.001 | 1,899 | 8 | 5 |
| 23 | 3/27 | 3/31 | 5 | 139.67 | 101 | 50 | 2.02 | <0.001 | 49,538 | 2 | 2 |
| 24 | 3/11 | 3/31 | 21 | 188.69 | 48 | 17 | 2.85 | <0.001 | 6,210 | 45 | 16 |

Note: Space-time clusters were identified using the spatial scan statistic with a Poisson model.

active space-time clusters at the end of May 20, 2020. From Fig 10 we can observe that in this period clusters continued to emerge in southern states and more clusters emerge in the mountain west. The previous cluster covering Nebraska and South Dakota has expanded into Iowa, North Dakota and Minneapolis. The metropolitan New York cluster has consolidated and the prior period mid-Atlantic cluster has consolidated to an emerging cluster around Philadelphia.

In this fourth period, using the sequence similarity-based clustering, we selected 10 clusters based on the elbow method evaluation (Fig 11). Fig 12 presents a map of these clusters and their temporal signatures. In this period, Cluster 8 which includes Miami, Chicago, Detroit, Los Angeles, Philadelphia and New York metropolitan counties is the fastest growing in term of cases. Clusters 7 and 9 start out with similar increases in cases but Cluster 7 members show a levelling off in early May relative to Cluster 9. Cluster 10 shows a delayed start but steady increase starting in early April. Cluster 5 shows a different trajectory in that it shows a much slower start to case accumulation but then exhibits a sharp increase starting in mid-April, increasing more rapidly than Clusters 10 and 7. Cluster 4 initially falls below the outside cluster "OC" group but then shows a sharp jump and more rapid accumulation. More detailed information on pairwise comparison of cluster numbers for the 4th study period from these two approaches can be found in S4 Table.

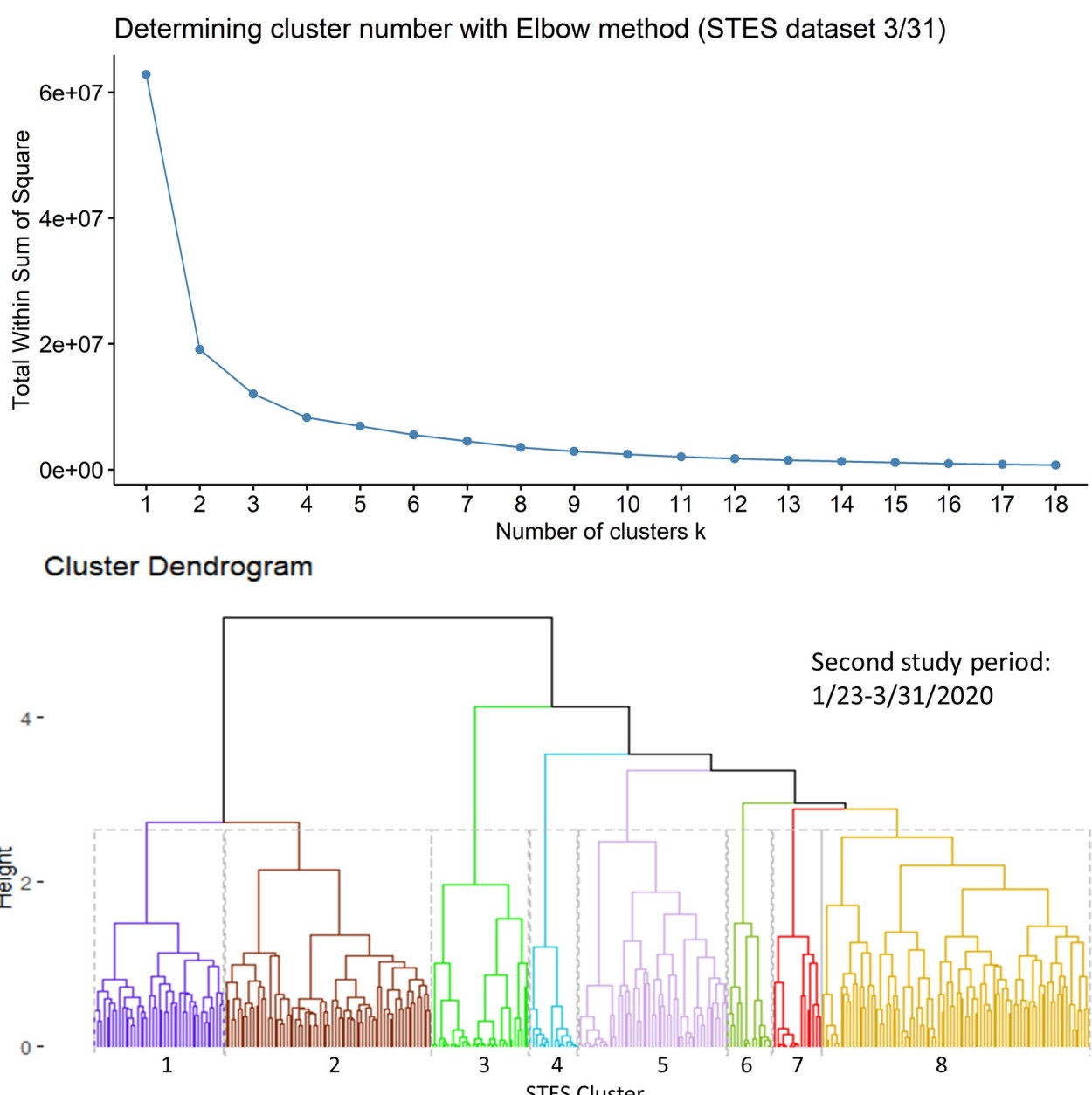

**Fig 5. Elbow method evaluation and hierarchical clustering results for the 2nd period.** Notice that the numberings and colors of STES clusters match with those of corresponding clusters on the map and the temporal trend graph in Fig 6.

## Discussion

For this study we compared two approaches for COVID-19 surveillance. In combination, the two approaches provide complementary views that can offer a more comprehensive picture of surveillance information to further aid public health analysis and monitoring. The space-time scan statistic identifies emerging clusters as locations where the observed number of cases most exceeds the expected number of cases in space-time based on the underlying population. This approach provokes questions of why the disease is emerging at such a location during a period of time. For disease progression, where the temporal pattern is equally important,

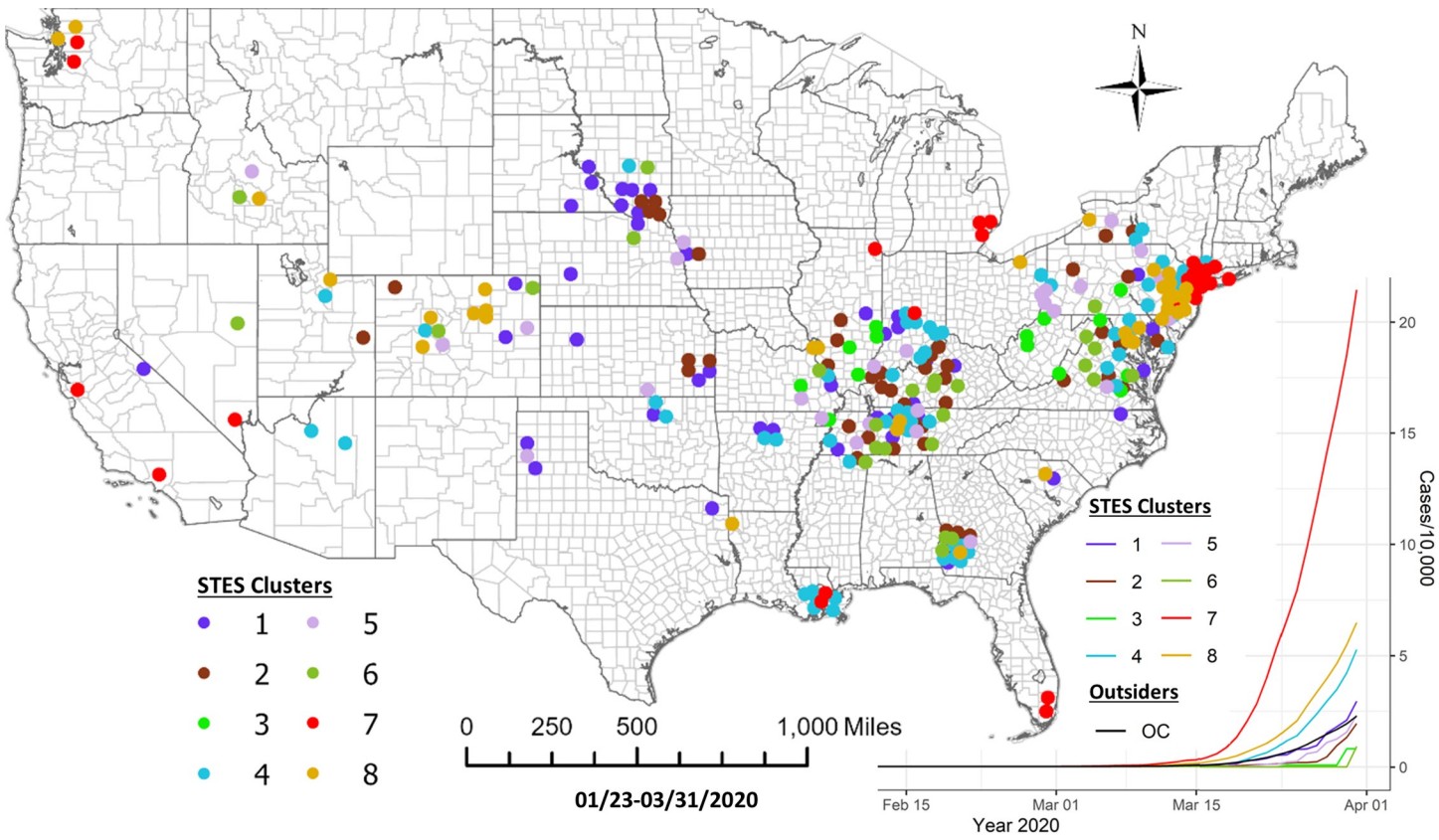

**Fig 6. Sequence similarity-based COVID-19 clusters along with average temporal trends at county level during 1/22/2020-3/31/2020.** This map includes the counties with higher relative risk (RR>1) contained in all the clusters detected by scan statistics in Fig 3. The average temporal trends of cumulative cases for STES clusters 1–8 on the map appear at the bottom right. Notice that the colors of STES clusters match with correspondingly colored dots on the map and with the colors of the STES cluster curves on the graph. OC includes all counties not included in the clusters.

similarity in the sequence of daily incidence rates adds valuable information as it points to locations where the disease is progressing in a similar fashion. This view provokes questions of why these sometimes spatially dispersed locations are behaving in a similar way.

An initial working hypothesis for the STES sequence similarity metric in an environmental monitoring context was that locations that are spatially close are more likely to exhibit similar event sequences. While this is born out in some instances in this pandemic context, we found that in all study periods, similar sequence patterns of COVID-19 cases can be quite spatially separated. This result suggests that spatial proximity is not always a driver of sequence similarity. It has been reported that socio-economic or demographic characteristics could explain the different transmission rates or patterns between communities and locations [30]. Because members of these clusters share similar temporal disease progressions, questions arise as to whether they share some similar underlying characteristics such as similar population density, similar populations at risk, similar changes in surveillance programs, or possibly similar intervention strategies at work.

Sequence similarity Cluster 3 in the first study period which covers the first appearance of COVID-19 in the US shows the earliest and fastest accumulating number of cases suggesting initial points of entry. As members of this cluster include Snohomish and King counties in Washington State, several California counties in the San Francisco Bay area, and Bronx, Kings, Queens, Wassau, and New York counties in New York state these do align with the known

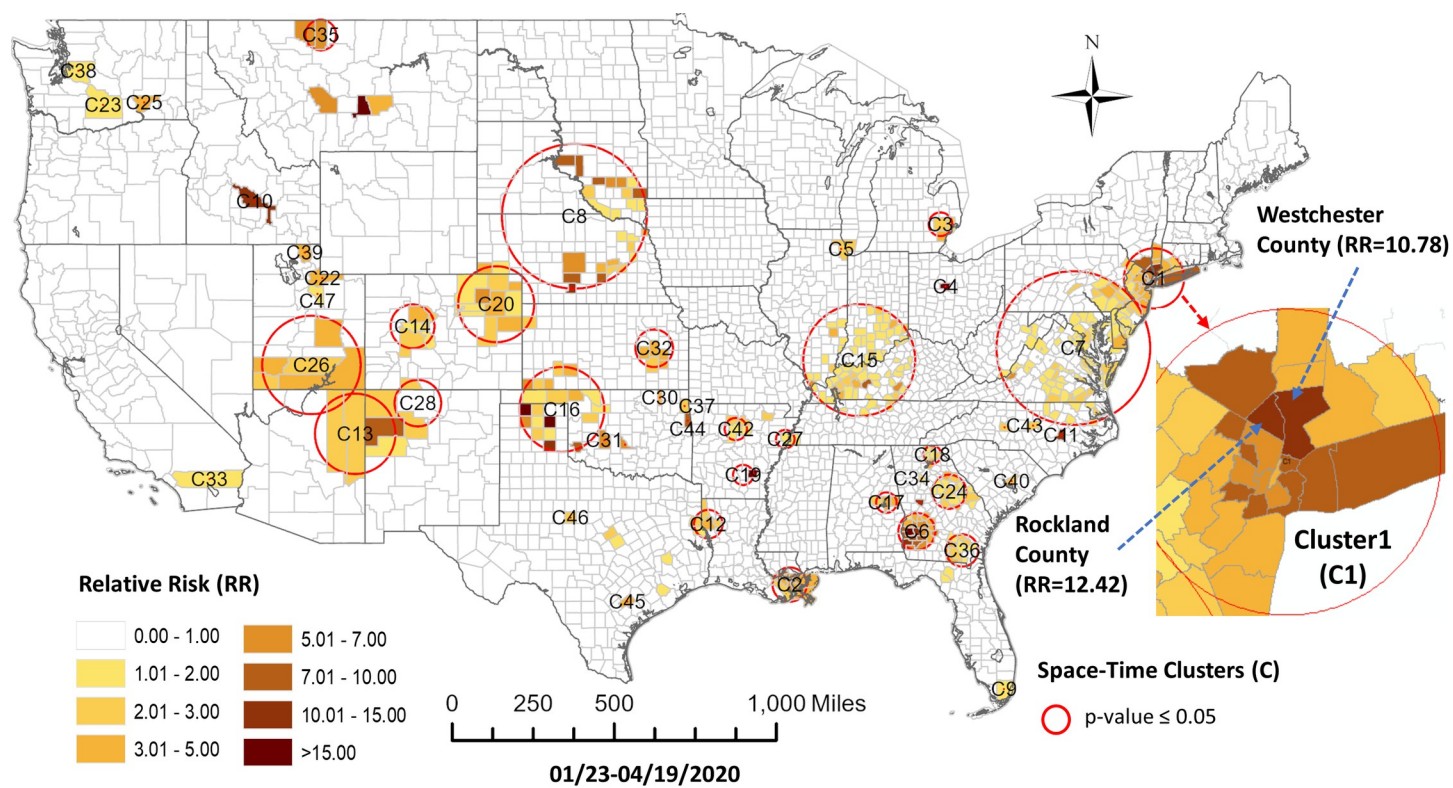

**Fig 7. COVID-19 space-time scan statistic detected hotspots in the United States at county level through 4/19/2020.**

entry points on the east and west coasts. Seemingly unusual members in this cluster are Johnson County Iowa; Kershaw County, South Carolina; Williamson, Tennessee; and Douglas, Nebraska. An interesting question is why this last subgroup of locations shares a similar profile with the coastal points of entry. Sequence similarity-based Cluster 2 in the first period is another interesting collection which is very spatially dispersed. Most of the members are rural communities that include Sheridan Wyoming, Davison South Dakota, Jackson Oklahoma, Hancock Indiana, Pitkin Colorado, Caddo Louisiana and Pierce Wisconsin. The temporal profile for this group is initially flat until mid-March at which point it shows a very rapid accumulation of cases. Such spatially dispersed cluster members that exhibit similar behaviours are targets for further investigation of potential contextual similarities. Of particular interest from epidemiological and health policy perspectives are spatially dispersed cluster members that exhibit similar flattening or decreasing patterns as these would be interesting to explore to understand if they have similar demographic characteristics or if they shared similar intervention measures.

We note that the sequence similarity clusters suggest some connections which are not conveyed by the scan statistic clusters. For example, in the third study period the scan statistic results indicate several new clusters. An examination of the sequence similarity clusters in this period indicate that several members of Cluster 10 were first nation or tribal reservations. In other words, several of the spatially dispersed reservations across the west showed a similar onset and progression in COVID-19 cases.

Another difference between the two approaches is that the sequence similarity-based clusters starting in the third period begin to show evidence of a spatial diffusion effect. For example, members of Cluster 8 with the earliest and fastest accumulating sequence similarity often

**Table 3. Attributes of prospective space-time clusters (hotspots) for COVID-19 from 1/23-4/19/2020 at the county level.**

| Cluster | Start Date | End Date | Duration (Days) | Radius (Km) | Observed Cases | Expected Cases | Relative Risk (RR) | p-value | Population at Risk | #County (total) | #County (RR>1) |
|---|---|---|---|---|---|---|---|---|---|---|---|
| 1 | 3/21 | 4/19 | 30 | 112.67 | 317,283 | 50,808 | 10.07 | <0.001 | 10,183,190 | 29 | 29 |
| 2 | 3/25 | 4/19 | 26 | 73.70 | 13,048 | 2,223 | 5.96 | <0.001 | 468,407 | 8 | 8 |
| 3 | 3/27 | 4/19 | 24 | 43.08 | 22,215 | 7,189 | 3.15 | <0.001 | 1,680,202 | 3 | 3 |
| 4 | 4/16 | 4/19 | 4 | 0.00 | 1,670 | 20 | 83.28 | <0.001 | 19,232 | 1 | 1 |
| 5 | 4/4 | 4/19 | 16 | 0.00 | 15,161 | 6,360 | 2.41 | <0.001 | 2,838,481 | 1 | 1 |
| 6 | 3/31 | 4/19 | 20 | 77.77 | 2,949 | 441 | 6.72 | <0.001 | 93,100 | 22 | 22 |
| 7 | 4/6 | 4/19 | 14 | 298.19 | 40,502 | 27,081 | 1.52 | <0.001 | 9,421,799 | 226 | 93 |
| 8 | 4/10 | 4/19 | 10 | 263.00 | 1,767 | 341 | 5.2 | <0.001 | 137,317 | 85 | 26 |
| 9 | 3/30 | 4/19 | 21 | 0.00 | 8,162 | 4,404 | 1.86 | <0.001 | 1,173,224 | 1 | 1 |
| 10 | 3/26 | 4/19 | 25 | 0.00 | 435 | 36 | 12.25 | <0.001 | 7,586 | 1 | 1 |
| 11 | 4/17 | 4/19 | 3 | 0.00 | 360 | 29 | 12.63 | <0.001 | 30,783 | 1 | 1 |
| 12 | 4/1 | 4/19 | 19 | 59.89 | 1,270 | 464 | 2.74 | <0.001 | 116,600 | 6 | 5 |
| 13 | 4/9 | 4/19 | 11 | 162.39 | 832 | 271 | 3.07 | <0.001 | 112,063 | 5 | 5 |
| 14 | 3/20 | 4/19 | 31 | 84.21 | 760 | 281 | 2.71 | <0.001 | 52,008 | 6 | 6 |
| 15 | 3/31 | 4/19 | 20 | 218.29 | 10,400 | 8,205 | 1.27 | <0.001 | 1,932,165 | 152 | 77 |
| 16 | 4/5 | 4/19 | 15 | 169.63 | 400 | 104 | 3.84 | <0.001 | 22,025 | 36 | 20 |
| 17 | 4/9 | 4/19 | 11 | 42.71 | 309 | 67 | 4.58 | <0.001 | 24,501 | 3 | 3 |
| 18 | 4/14 | 4/19 | 6 | 36.59 | 428 | 142 | 3.02 | <0.001 | 97,393 | 6 | 6 |
| 19 | 4/13 | 4/19 | 7 | 41.53 | 100 | 6 | 16.58 | <0.001 | 2,434 | 2 | 1 |
| 20 | 4/9 | 4/19 | 11 | 144.34 | 2,800 | 1,943 | 1.44 | <0.001 | 999,773 | 20 | 14 |
| 21 | 4/14 | 4/19 | 6 | 0.00 | 109 | 10 | 10.73 | <0.001 | 5,683 | 1 | 1 |
| 22 | 3/20 | 4/19 | 31 | 0.00 | 299 | 88 | 3.41 | <0.001 | 16,762 | 1 | 1 |
| 23 | 4/2 | 4/19 | 18 | 0.00 | 643 | 349 | 1.85 | <0.001 | 94,077 | 1 | 1 |
| 24 | 4/7 | 4/19 | 13 | 70.67 | 348 | 179 | 1.95 | <0.001 | 41,649 | 17 | 14 |
| 25 | 4/15 | 4/19 | 5 | 0.00 | 123 | 37 | 3.35 | <0.001 | 29,216 | 1 | 1 |
| 26 | 4/17 | 4/19 | 3 | 192.58 | 142 | 51 | 2.8 | <0.001 | 50,741 | 11 | 6 |
| 27 | 4/18 | 4/19 | 2 | 37.48 | 298 | 152 | 1.96 | <0.001 | 386,360 | 2 | 2 |
| 28 | 4/3 | 4/19 | 17 | 92.71 | 301 | 156 | 1.93 | <0.001 | 41,584 | 5 | 3 |
| 29 | 4/11 | 4/19 | 9 | 0.00 | 173 | 77 | 2.25 | <0.001 | 41,981 | 1 | 1 |
| 30 | 4/11 | 4/19 | 9 | 0.00 | 83 | 24 | 3.48 | <0.001 | 14,638 | 1 | 1 |
| 31 | 4/15 | 4/19 | 5 | 0.00 | 41 | 7 | 6.16 | <0.001 | 3,595 | 1 | 1 |
| 32 | 4/15 | 4/19 | 5 | 72.81 | 57 | 13 | 4.29 | <0.001 | 10,680 | 8 | 5 |
| 33 | 4/14 | 4/19 | 6 | 0.00 | 1,019 | 763 | 1.34 | <0.001 | 926,455 | 1 | 1 |
| 34 | 4/13 | 4/19 | 7 | 0.00 | 583 | 410 | 1.42 | <0.001 | 336,507 | 1 | 1 |
| 35 | 3/28 | 4/19 | 23 | 50.34 | 32 | 5 | 6.04 | <0.001 | 888 | 2 | 2 |
| 36 | 4/2 | 4/19 | 18 | 68.61 | 253 | 149 | 1.7 | <0.001 | 28,897 | 10 | 9 |
| 37 | 4/12 | 4/19 | 8 | 0.00 | 59 | 20 | 2.96 | <0.001 | 8,797 | 1 | 1 |
| 38 | 4/18 | 4/19 | 2 | 0.00 | 272 | 174 | 1.56 | <0.001 | 1,139,191 | 1 | 1 |
| 39 | 4/17 | 4/19 | 3 | 0.00 | 37 | 10 | 3.74 | <0.001 | 27,699 | 1 | 1 |
| 40 | 3/29 | 4/19 | 22 | 0.00 | 105 | 52 | 2.02 | <0.001 | 9,587 | 1 | 1 |
| 41 | 4/18 | 4/19 | 2 | 0.00 | 20 | 3 | 6.4 | <0.001 | 7,819 | 1 | 1 |
| 42 | 3/23 | 4/19 | 28 | 44.85 | 112 | 58 | 1.94 | <0.001 | 9,320 | 5 | 5 |
| 43 | 4/11 | 4/19 | 9 | 0.00 | 93 | 46 | 2.02 | <0.001 | 17,771 | 1 | 1 |
| 44 | 4/18 | 4/19 | 2 | 0.00 | 14 | 2 | 8.17 | 0.002 | 3,531 | 1 | 1 |
| 45 | 4/14 | 4/19 | 6 | 0.00 | 22 | 5 | 4.71 | 0.003 | 2,749 | 1 | 1 |
| 46 | 4/18 | 4/19 | 2 | 0.00 | 53 | 21 | 2.49 | 0.003 | 31,371 | 1 | 1 |

*(Continued)*

**Table 3.** (Continued)

| Cluster | Start Date | End Date | Duration (Days) | Radius (Km) | Observed Cases | Expected Cases | Relative Risk (RR) | p-value | Population at Risk | #County (total) | #County (RR>1) |
|---|---|---|---|---|---|---|---|---|---|---|---|
| 47 | 3/24 | 4/19 | 27 | 0.00 | 102 | 55 | 1.85 | 0.006 | 10,847 | 1 | 1 |

Note: Space-time clusters were identified using the spatial scan statistic with a Poisson model.

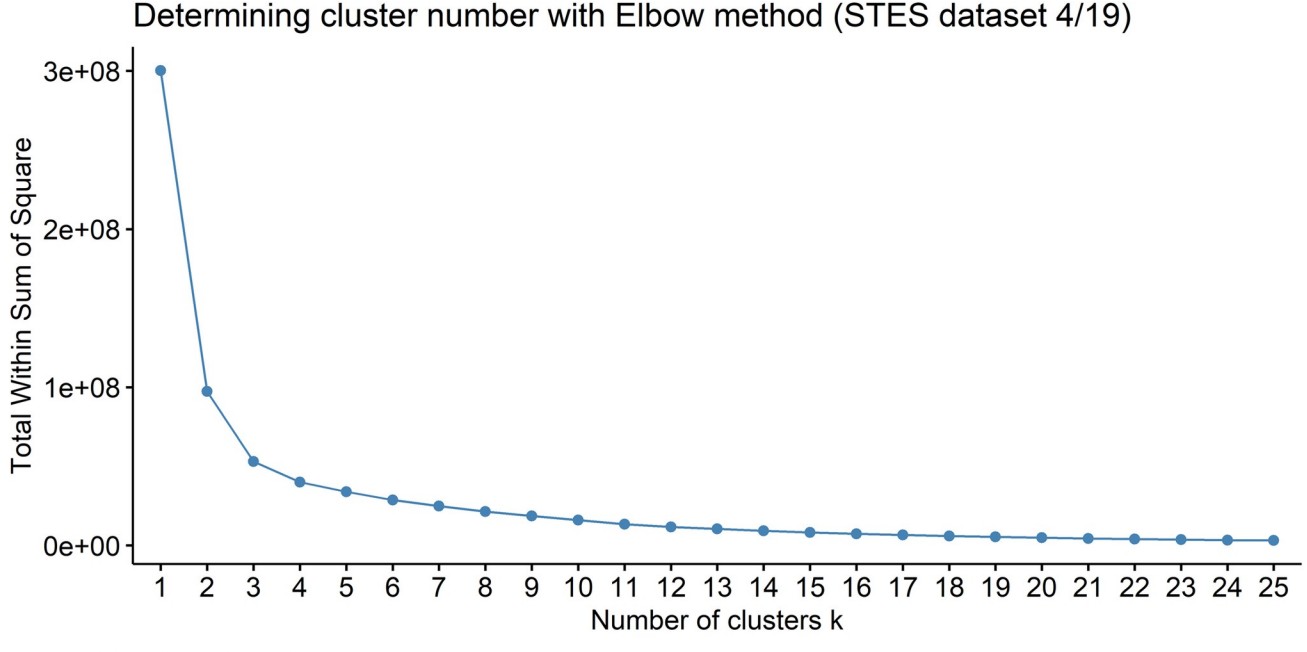

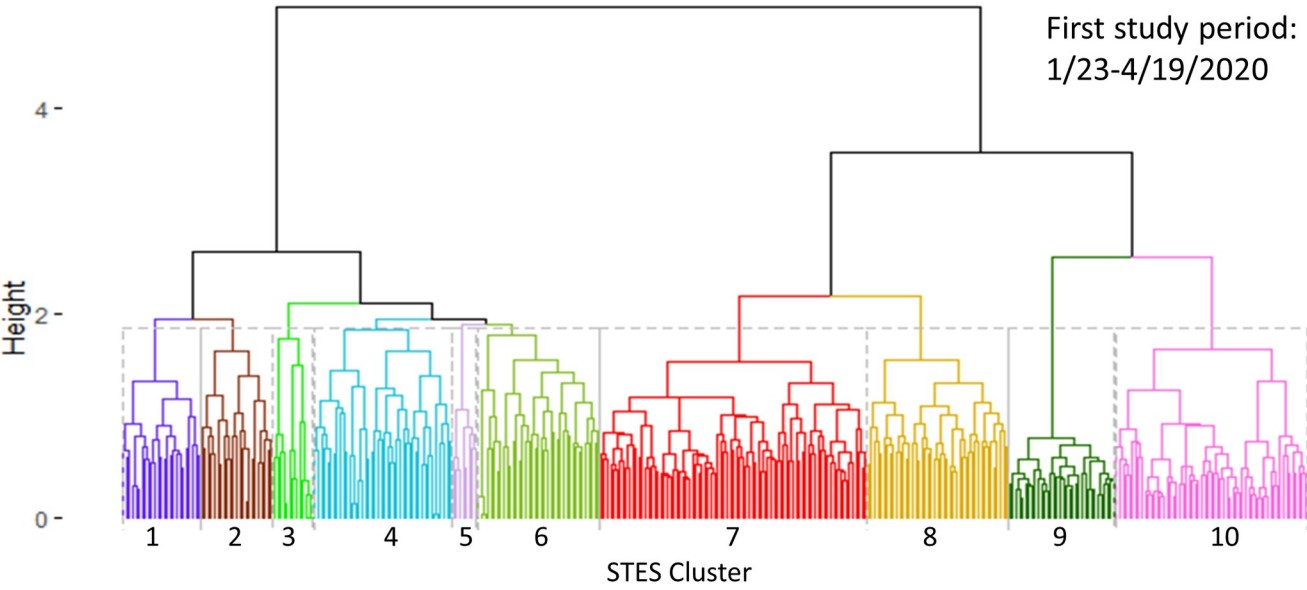

**Fig 8. Elbow method evaluation and hierarchical clustering results for the 3$^{rd}$ period.** Notice that the numberings and colors of STES clusters match with those of corresponding clusters on the map and the temporal trend graph in Fig 9.

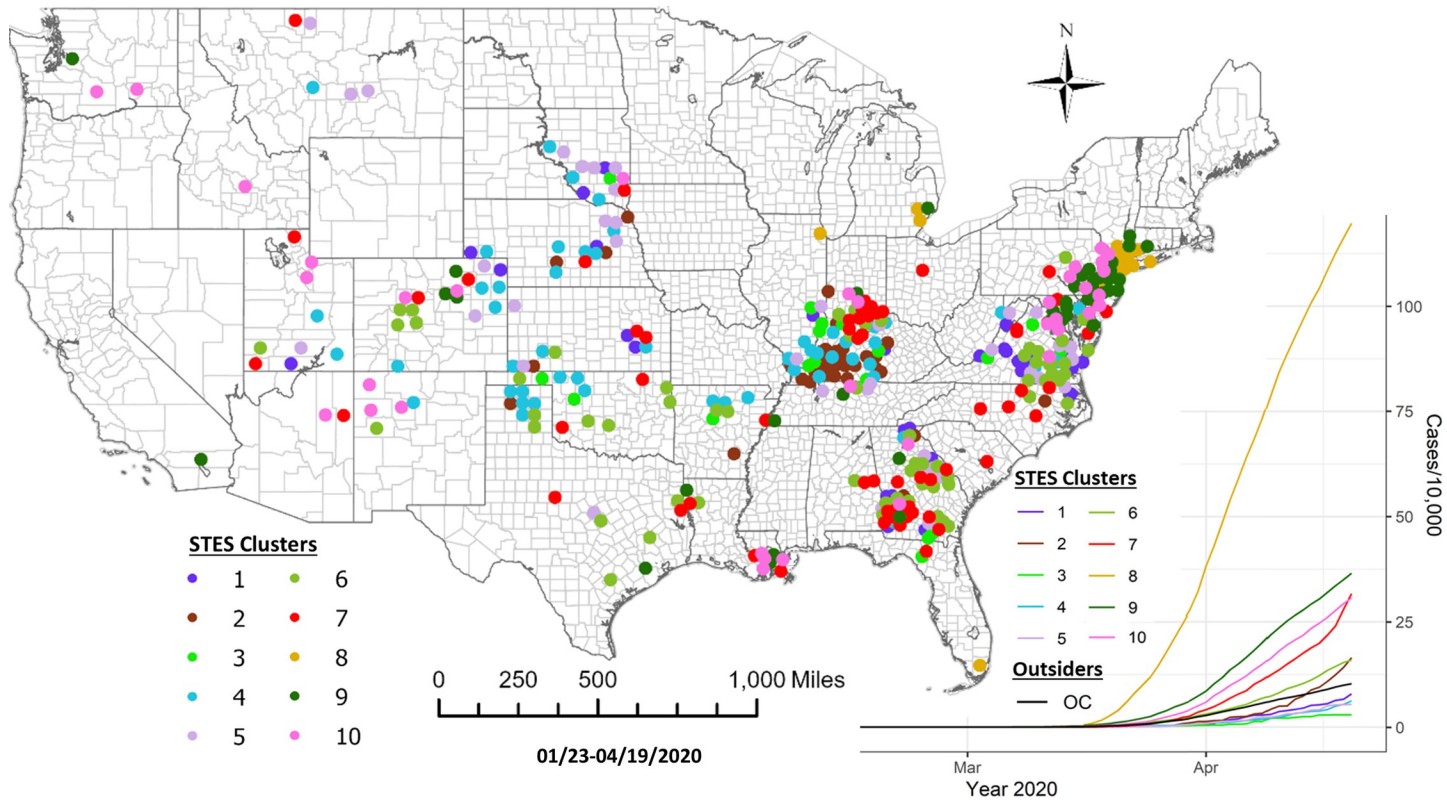

**Fig 9. Sequence similarity-based COVID-19 emerging clusters along with average temporal trends at county level during 1/22/-4/19/2020.** This map includes the counties with higher relative risk (RR>) contained in all the clusters detected by scan statistics in Fig 5. The average temporal trends of cumulative cases for STES clusters 1–10 on the map appear at the bottom right. Notice that the colors of STES clusters match with correspondingly colored dots on the map and with the colors of the STES cluster curves on the graph. OC includes all counties not included in the clusters.

**Table 4. Attributes of prospective space-time clusters (hotspots) for COVID-19 from 1/23-5/20/2020 at the county level.**

| Cluster | Start Date | End Date | Duration (Days) | Radius (Km) | Observed Cases | Expected Cases | Relative Risk (RR) | p-value | Population at Risk | #County (total) | #County (RR>1) |
|---|---|---|---|---|---|---|---|---|---|---|---|
| 1 | 3/23 | 5/20 | 59 | 126.60 | 516,153 | 128,515 | 5.51 | <0.001 | 15,225,284 | 35 | 35 |
| 2 | 4/7 | 5/20 | 44 | 55.64 | 77,744 | 30,138 | 2.66 | <0.001 | 5,000,478 | 5 | 5 |
| 3 | 4/12 | 5/20 | 39 | 332.91 | 14,779 | 3,116 | 4.78 | <0.001 | 411,108 | 155 | 109 |
| 4 | 4/17 | 5/20 | 34 | 103.56 | 41,285 | 18,966 | 2.21 | <0.001 | 3,575,889 | 42 | 25 |
| 5 | 4/20 | 5/20 | 31 | 215.21 | 7,183 | 749 | 9.63 | <0.001 | 111,251 | 47 | 35 |
| 6 | 3/23 | 5/20 | 59 | 73.70 | 16,614 | 5,499 | 3.04 | <0.001 | 625,641 | 8 | 8 |
| 7 | 3/26 | 5/20 | 56 | 43.08 | 34,409 | 18,624 | 1.87 | <0.001 | 2,253,493 | 3 | 3 |
| 8 | 4/29 | 5/20 | 22 | 0.00 | 1,336 | 16 | 81.34 | <0.001 | 3,508 | 1 | 1 |
| 9 | 4/13 | 5/20 | 38 | 0.00 | 2,487 | 206 | 12.07 | <0.001 | 30,632 | 1 | 1 |
| 10 | 4/9 | 5/20 | 42 | 191.99 | 5,571 | 1,339 | 4.17 | <0.001 | 184,726 | 6 | 6 |
| 11 | 4/15 | 5/20 | 36 | 0.00 | 1,952 | 175 | 11.15 | <0.001 | 25,544 | 1 | 1 |
| 12 | 3/24 | 5/20 | 58 | 77.77 | 4,684 | 1,282 | 3.66 | <0.001 | 134,101 | 22 | 22 |
| 13 | 4/13 | 5/20 | 38 | 0.00 | 955 | 36 | 26.75 | <0.001 | 4,378 | 1 | 1 |
| 14 | 4/15 | 5/20 | 36 | 114.37 | 3,799 | 1,339 | 2.84 | <0.001 | 187,231 | 21 | 21 |
| 15 | 4/23 | 5/20 | 28 | 0.00 | 598 | 21 | 28.96 | <0.001 | 3,038 | 1 | 1 |
| 16 | 5/12 | 5/20 | 9 | 0.00 | 344 | 3 | 114.45 | <0.001 | 1,002 | 1 | 1 |
| 17 | 4/14 | 5/20 | 37 | 36.59 | 2,623 | 962 | 2.73 | <0.001 | 150,923 | 6 | 5 |
| 18 | 4/24 | 5/20 | 27 | 42.39 | 1,579 | 458 | 3.45 | <0.001 | 77,989 | 7 | 7 |

*(Continued)*

**Table 4.** (Continued)

| Cluster | Start Date | End Date | Duration (Days) | Radius (Km) | Observed Cases | Expected Cases | Relative Risk (RR) | p-value | Population at Risk | #County (total) | #County (RR>1) |
|---|---|---|---|---|---|---|---|---|---|---|---|
| 19 | 4/30 | 5/20 | 21 | 0.00 | 1,436 | 451 | 3.18 | <0.001 | 134,923 | 1 | 1 |
| 20 | 5/3 | 5/20 | 18 | 0.00 | 191 | 4 | 44.47 | <0.001 | 772 | 1 | 1 |
| 21 | 3/23 | 5/20 | 59 | 47.10 | 519 | 87 | 5.99 | <0.001 | 9,665 | 2 | 2 |
| 22 | 4/28 | 5/20 | 23 | 45.28 | 436 | 77 | 5.66 | <0.001 | 13,095 | 3 | 3 |
| 23 | 5/10 | 5/20 | 11 | 29.09 | 221 | 15 | 14.38 | <0.001 | 3,235 | 3 | 3 |
| 24 | 5/10 | 5/20 | 11 | 0.00 | 257 | 24 | 10.91 | <0.001 | 7,981 | 1 | 1 |
| 25 | 4/30 | 5/20 | 21 | 0.00 | 354 | 56 | 6.27 | <0.001 | 11,332 | 1 | 1 |
| 26 | 5/6 | 5/20 | 15 | 0.00 | 994 | 383 | 2.6 | <0.001 | 202,613 | 1 | 1 |
| 27 | 4/1 | 5/20 | 50 | 136.56 | 5,564 | 3,846 | 1.45 | <0.001 | 449,669 | 30 | 22 |
| 28 | 5/7 | 5/20 | 14 | 0.00 | 566 | 155 | 3.65 | <0.001 | 71,572 | 1 | 1 |
| 29 | 5/2 | 5/20 | 19 | 31.84 | 510 | 133 | 3.83 | <0.001 | 20,764 | 4 | 4 |
| 30 | 4/19 | 5/20 | 32 | 192.58 | 305 | 51 | 6.02 | <0.001 | 40,867 | 11 | 2 |
| 31 | 3/30 | 5/20 | 52 | 0.00 | 14,842 | 12,107 | 1.23 | <0.001 | 1,575,369 | 1 | 1 |
| 32 | 4/21 | 5/20 | 30 | 0.00 | 517 | 144 | 3.6 | <0.001 | 25,141 | 1 | 1 |
| 33 | 5/11 | 5/20 | 10 | 0.00 | 248 | 37 | 6.71 | <0.001 | 24,329 | 1 | 1 |
| 34 | 5/12 | 5/20 | 9 | 45.71 | 262 | 47 | 5.53 | <0.001 | 32,224 | 3 | 1 |
| 35 | 4/27 | 5/20 | 24 | 0.00 | 153 | 16 | 9.6 | <0.001 | 2,345 | 1 | 1 |
| 36 | 4/29 | 5/20 | 22 | 37.68 | 576 | 218 | 2.65 | <0.001 | 48,225 | 2 | 2 |
| 37 | 4/2 | 5/20 | 49 | 42.71 | 704 | 312 | 2.25 | <0.001 | 36,636 | 3 | 3 |
| 38 | 5/8 | 5/20 | 13 | 0.00 | 164 | 24 | 6.95 | <0.001 | 5,473 | 1 | 1 |
| 39 | 5/19 | 5/20 | 2 | 0.00 | 2,437 | 1,721 | 1.42 | <0.001 | 6,453,712 | 1 | 1 |
| 40 | 5/15 | 5/20 | 6 | 0.00 | 60 | 3 | 21 | <0.001 | 841 | 1 | 1 |
| 41 | 5/6 | 5/20 | 15 | 29.36 | 112 | 17 | 6.41 | <0.001 | 4,070 | 2 | 2 |
| 42 | 5/10 | 5/20 | 11 | 45.67 | 150 | 32 | 4.62 | <0.001 | 8,166 | 2 | 2 |
| 43 | 4/6 | 5/20 | 45 | 30.61 | 309 | 116 | 2.67 | <0.001 | 13,014 | 3 | 3 |
| 44 | 4/18 | 5/20 | 33 | 0.00 | 519 | 257 | 2.02 | <0.001 | 42,288 | 1 | 1 |
| 45 | 5/7 | 5/20 | 14 | 0.00 | 105 | 20 | 5.2 | <0.001 | 5,939 | 1 | 1 |
| 46 | 4/25 | 5/20 | 26 | 99.90 | 124 | 29 | 4.23 | <0.001 | 4,072 | 15 | 6 |
| 47 | 4/20 | 5/20 | 31 | 30.03 | 288 | 124 | 2.33 | <0.001 | 22,341 | 3 | 2 |
| 48 | 3/23 | 5/20 | 59 | 77.39 | 581 | 342 | 1.7 | <0.001 | 39,119 | 4 | 2 |
| 49 | 5/13 | 5/20 | 8 | 106.86 | 270 | 121 | 2.24 | <0.001 | 83,127 | 2 | 2 |
| 50 | 3/29 | 5/20 | 53 | 0.00 | 291 | 139 | 2.1 | <0.001 | 15,029 | 1 | 1 |
| 51 | 4/22 | 5/20 | 29 | 26.90 | 155 | 55 | 2.83 | <0.001 | 8,779 | 2 | 2 |
| 52 | 4/7 | 5/20 | 44 | 46.15 | 317 | 165 | 1.92 | <0.001 | 18,980 | 6 | 6 |
| 53 | 5/2 | 5/20 | 19 | 0.00 | 103 | 33 | 3.16 | <0.001 | 7,699 | 1 | 1 |
| 54 | 4/1 | 5/20 | 50 | 53.19 | 83 | 22 | 3.7 | <0.001 | 2,198 | 3 | 3 |
| 55 | 4/14 | 5/20 | 37 | 27.39 | 68 | 16 | 4.25 | <0.001 | 1,791 | 2 | 2 |
| 56 | 4/23 | 5/20 | 28 | 0.00 | 156 | 65 | 2.4 | <0.001 | 10,718 | 1 | 1 |
| 57 | 4/13 | 5/20 | 38 | 21.26 | 248 | 128 | 1.93 | <0.001 | 19,711 | 2 | 2 |
| 58 | 4/27 | 5/20 | 24 | 0.00 | 30 | 3 | 10.24 | <0.001 | 323 | 1 | 1 |
| 59 | 5/18 | 5/20 | 3 | 0.00 | 49 | 9 | 5.29 | <0.001 | 15,448 | 1 | 1 |
| 60 | 4/17 | 5/20 | 34 | 0.00 | 107 | 39 | 2.73 | <0.001 | 8,405 | 1 | 1 |
| 61 | 4/18 | 5/20 | 33 | 72.28 | 534 | 354 | 1.51 | <0.001 | 58,978 | 7 | 5 |
| 62 | 4/21 | 5/20 | 30 | 140.99 | 233 | 125 | 1.87 | <0.001 | 26,408 | 6 | 4 |
| 63 | 4/29 | 5/20 | 22 | 0.00 | 234 | 126 | 1.85 | <0.001 | 30,406 | 1 | 1 |
| 64 | 4/22 | 5/20 | 29 | 0.00 | 115 | 47 | 2.43 | <0.001 | 6,538 | 1 | 1 |
| 65 | 5/19 | 5/20 | 2 | 0.00 | 21 | 2 | 12.77 | <0.001 | 4,032 | 1 | 1 |
| 66 | 5/5 | 5/20 | 16 | 92.43 | 1,039 | 796 | 1.3 | <0.001 | 286,527 | 2 | 2 |
| 67 | 4/19 | 5/20 | 32 | 0.00 | 115 | 49 | 2.37 | <0.001 | 10,204 | 1 | 1 |
| 68 | 5/8 | 5/20 | 13 | 0.00 | 192 | 101 | 1.9 | <0.001 | 45,852 | 1 | 1 |

*(Continued)*

**Table 4.** (Continued)

| Cluster | Start Date | End Date | Duration (Days) | Radius (Km) | Observed Cases | Expected Cases | Relative Risk (RR) | p-value | Population at Risk | #County (total) | #County (RR>1) |
|---------|-----------|----------|-----------------|-------------|----------------|----------------|--------------------|---------|--------------------|-----------------|----------------|
| 69 | 5/12 | 5/20 | 9 | 0.00 | 30 | 4 | 6.87 | <0.001 | 771 | 1 | 1 |
| 70 | 5/17 | 5/20 | 4 | 0.00 | 123 | 55 | 2.23 | <0.001 | 78,471 | 1 | 1 |
| 71 | 4/29 | 5/20 | 22 | 0.00 | 156 | 79 | 1.97 | <0.001 | 17,303 | 1 | 1 |
| 72 | 3/28 | 5/20 | 54 | 50.34 | 32 | 6 | 5.44 | <0.001 | 656 | 2 | 2 |
| 73 | 5/7 | 5/20 | 14 | 80.26 | 106 | 46 | 2.28 | <0.001 | 12,240 | 5 | 4 |
| 74 | 4/14 | 5/20 | 37 | 47.62 | 115 | 53 | 2.15 | <0.001 | 7,305 | 3 | 3 |
| 75 | 4/9 | 5/20 | 42 | 35.79 | 123 | 59 | 2.09 | <0.001 | 6,343 | 2 | 2 |
| 76 | 4/20 | 5/20 | 31 | 0.00 | 134 | 68 | 1.98 | <0.001 | 11,760 | 1 | 1 |
| 77 | 4/28 | 5/20 | 23 | 195.74 | 281 | 184 | 1.53 | <0.001 | 48,676 | 9 | 4 |
| 78 | 4/16 | 5/20 | 35 | 27.34 | 243 | 154 | 1.57 | <0.001 | 22,877 | 3 | 2 |
| 79 | 4/15 | 5/20 | 36 | 0.00 | 116 | 59 | 1.96 | <0.001 | 7,734 | 1 | 1 |
| 80 | 4/9 | 5/20 | 42 | 56.31 | 478 | 350 | 1.36 | <0.001 | 49,008 | 2 | 1 |
| 81 | 5/18 | 5/20 | 3 | 93.41 | 130 | 70 | 1.86 | <0.001 | 180,113 | 8 | 2 |
| 82 | 4/17 | 5/20 | 34 | 0.00 | 37 | 11 | 3.37 | <0.001 | 20,483 | 1 | 1 |
| 83 | 4/10 | 5/20 | 41 | 30.49 | 135 | 76 | 1.78 | <0.001 | 9,851 | 2 | 2 |
| 84 | 5/14 | 5/20 | 7 | 0.00 | 125 | 69 | 1.82 | <0.001 | 43,779 | 1 | 1 |
| 85 | 5/12 | 5/20 | 9 | 27.78 | 87 | 44 | 1.97 | 0.004 | 16,827 | 2 | 1 |
| 86 | 5/3 | 5/20 | 18 | 0.00 | 20 | 4 | 4.6 | 0.013 | 670 | 1 | 1 |
| 87 | 5/19 | 5/20 | 2 | 80.43 | 28 | 8 | 3.38 | 0.019 | 55,557 | 12 | 2 |

Note: Space-time clusters were identified using the spatial scan statistic with a Poisson model.

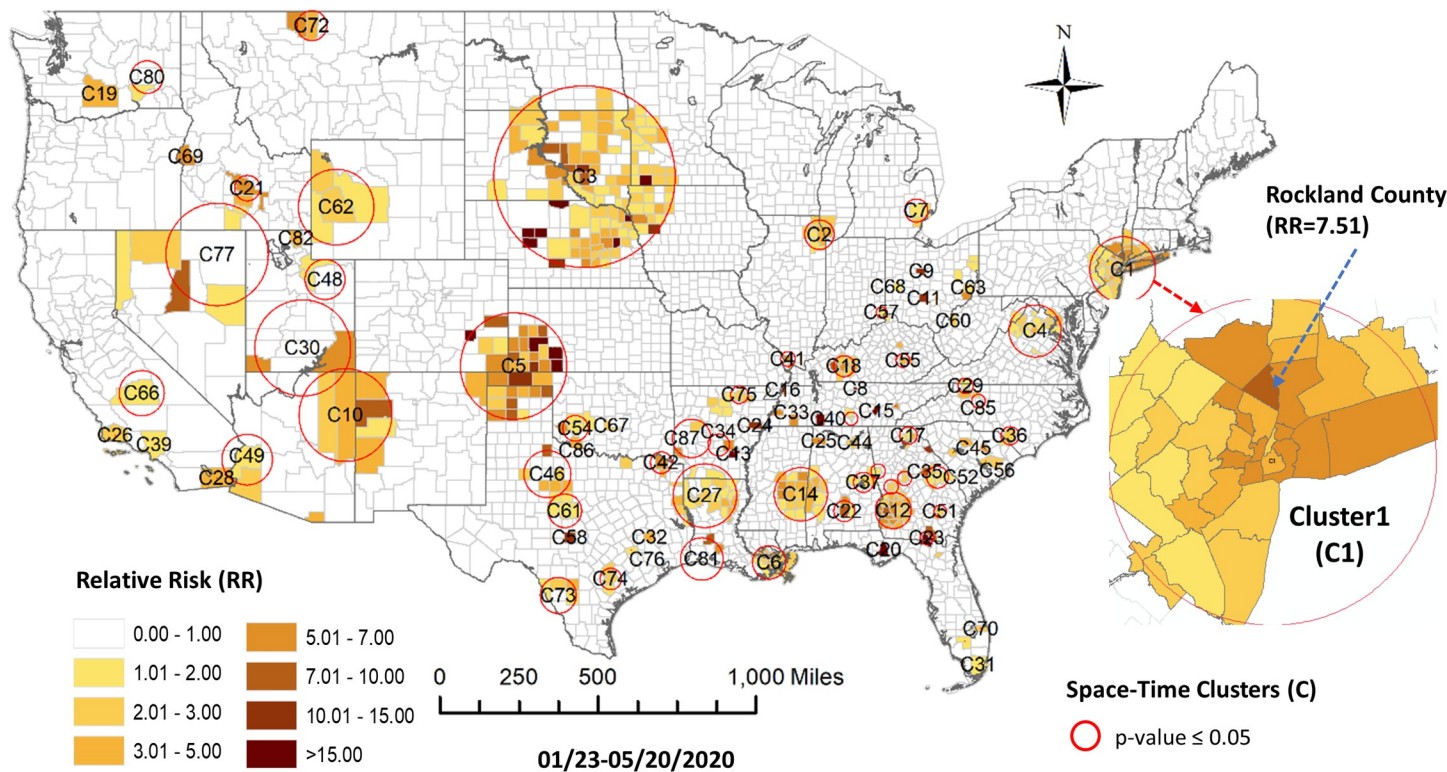

**Fig 10. Prospective space-time scan statistic detected clusters of COVID-19 incidents during the study period of 1/22/2020-5/20/2020.**

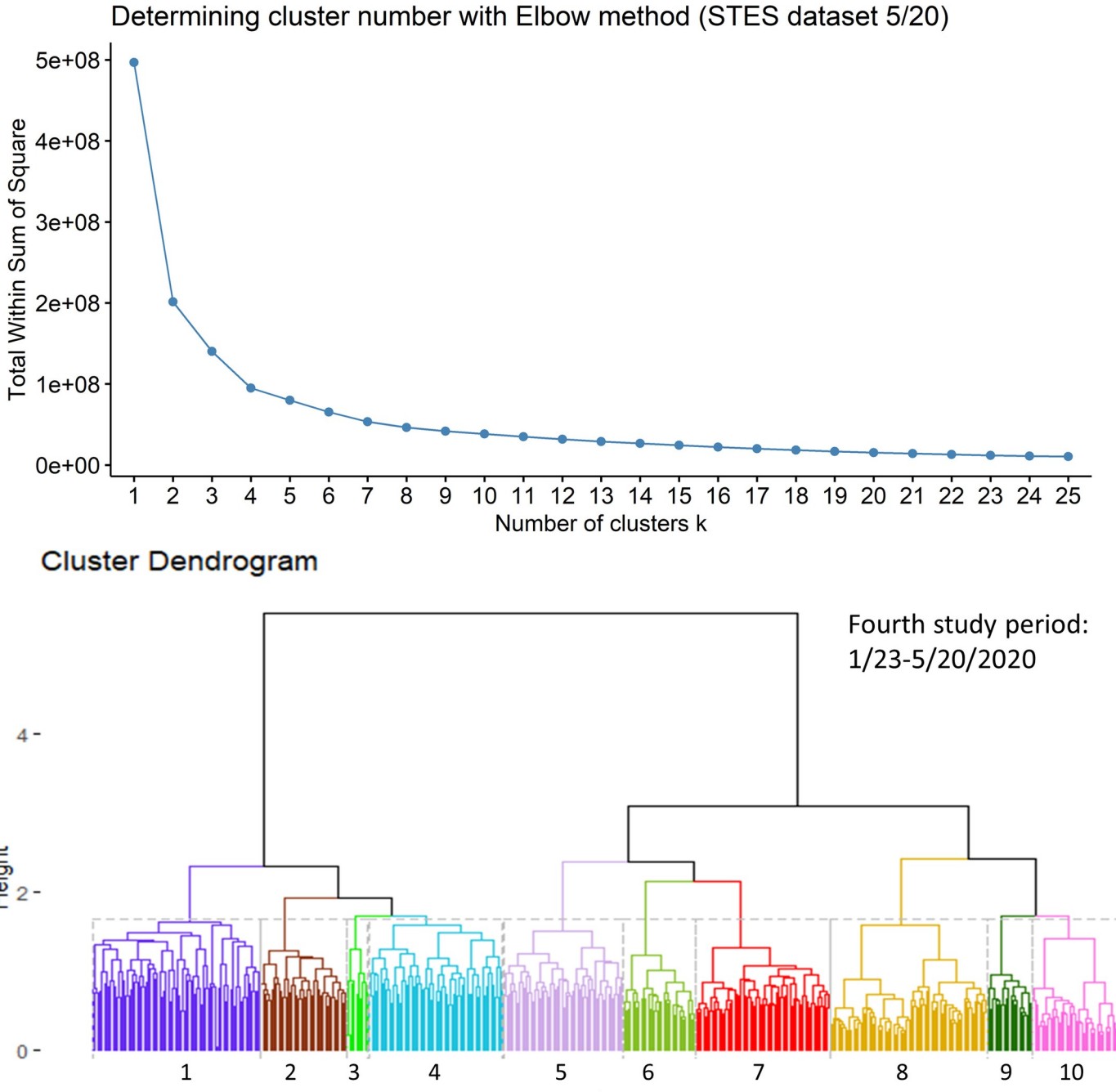

**Fig 11. Elbow method evaluation and hierarchical clustering results for the 4ᵗʰ period.** Notice that the numberings and colors of STES clusters match with those of corresponding clusters on the map and the temporal trend graph in Fig 12.

appear to be surrounded by or in close spatial association with the next closest lagging group, Cluster 9. A similar pattern appears between Cluster 8 and Cluster 9 members in the fourth study period.

Recent research has pointed to different continents of origin for the introduction of COVID-19 into the US [31, 32]. Genomic epidemiology research supports the belief that

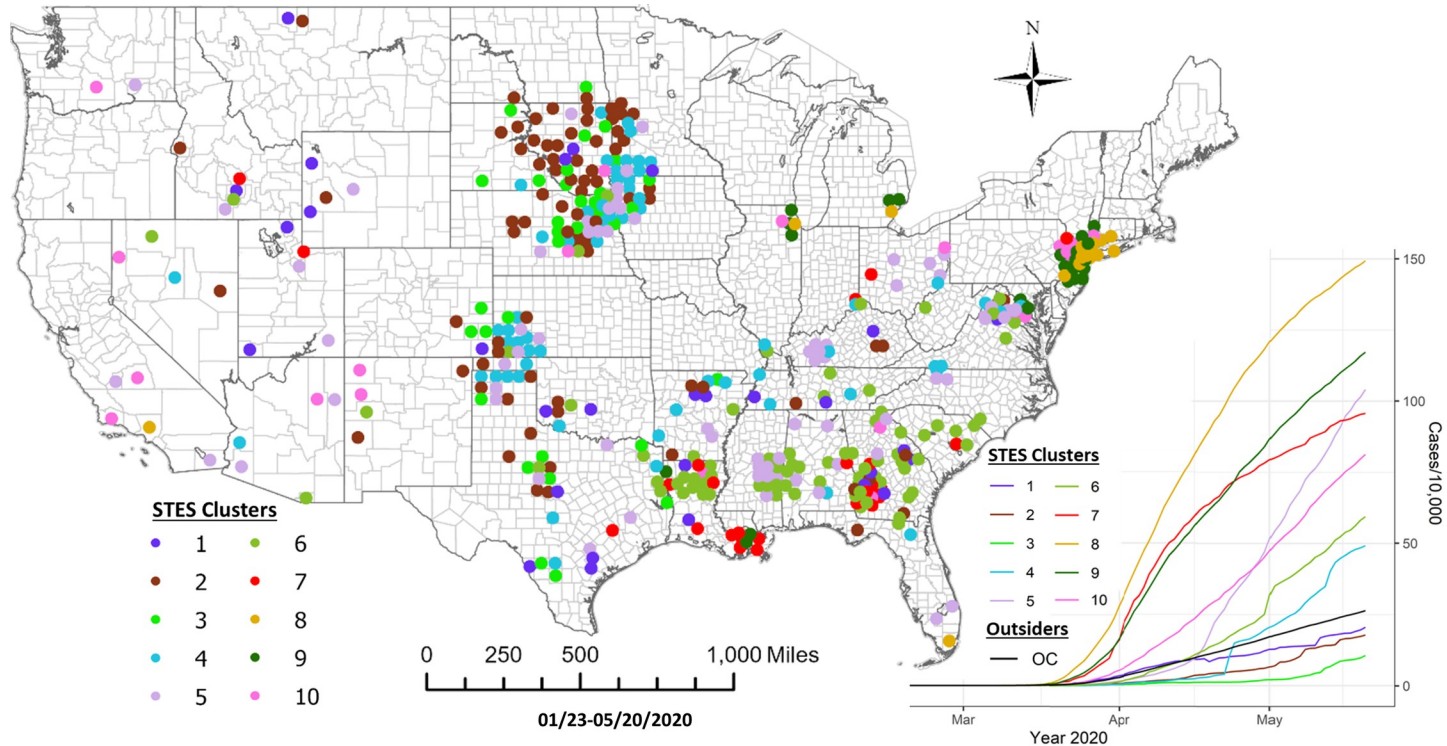

**Fig 12. Sequence similarity-based COVID-19 clusters along with average temporal trends at county level during 1/22/-5/20/2020.** This map includes the counties with higher relative risk (RR>1) contained in all the clusters detected by scan statistics in Fig 10. The average temporal trends of cumulative cases for STES clusters 1–10 on the map appear at the bottom right. Notice that the colors of STES clusters match with correspondingly colored dots on the map and with the colors of the STES cluster curves on the graph. OC includes all counties not included in the clusters.

isolates from China primarily seeded the original COVID-19 outbreak on the US West Coast and that European isolates seeded the pandemic in New York (and the US East Coast) [33]. Given some connectivity suggested by the sequence similarity based approach there may exist opportunities for productive combination with phylogenetic tracing and transmission pathway studies [34].

We recognize that both approaches can be impacted by limitations in data collection. Several publications have noted reporting lags although these are most problematic with respect to death reports rather than daily reported case counts [35–38]. There is clearly the potential for inaccuracies in data collection covering many different jurisdictions. If for example, reports of new cases are delayed by a day or two from a jurisdiction this could potentially change the similarity in the sequences of county daily case counts. However, given the length of the study periods here we expect lags of one to two days to have minor impact.

## Supporting information

**S1 Table. Comparison of space-time clusters from SaTScan and STES based hierarchical clustering with the dataset from 1/23-3-13/2020.** This table is merged through FIPS of US counties, and also includes other selected output parameters from SaTScan such as p-values, LOC_RR (location or county relative risk), CLU_RR (cluster relative risk), LOC_LAT (location latitude), LOC_LONG (location longitude).
(XLSX)

**S2 Table. Comparison of space-time clusters from SaTScan and STES based hierarchical clustering with the dataset from 1/23-3-31/2020.** This table is merged through FIPS of US counties, and also includes other selected output parameters from SaTScan such as p-values, LOC_RR (location or county relative risk), CLU_RR (cluster relative risk), LOC_LAT (location latitude), LOC_LONG (location longitude).
(XLSX)

**S3 Table. Comparison of space-time clusters from SaTScan and STES based hierarchical clustering with the dataset from 1/23-4-19/2020.** This table is merged through FIPS of US counties, and also includes other selected output parameters from SaTScan such as p-values, LOC_RR (location or county relative risk), CLU_RR (cluster relative risk), LOC_LAT (location latitude), LOC_LONG (location longitude).
(XLSX)

**S4 Table. Comparison of space-time clusters from SaTScan and STES based hierarchical clustering with the dataset from 1/23-5-20/2020.** This table is merged through FIPS of US counties, and also includes other selected output parameters from SaTScan such as p-values, LOC_RR (location or county relative risk), CLU_RR (cluster relative risk), LOC_LAT (location latitude), LOC_LONG (location longitude).
(XLSX)

**S5 Table. The minimal data set underlying the results described in this manuscript.**
(CSV)

## Author Contributions

**Conceptualization:** Fuyu Xu, Kate Beard.

**Data curation:** Fuyu Xu.

**Formal analysis:** Fuyu Xu, Kate Beard.

**Investigation:** Fuyu Xu, Kate Beard.

**Methodology:** Fuyu Xu.

**Software:** Fuyu Xu, Kate Beard.

**Supervision:** Kate Beard.

**Validation:** Fuyu Xu, Kate Beard.

**Visualization:** Fuyu Xu.

**Writing – original draft:** Fuyu Xu.

**Writing – review & editing:** Fuyu Xu, Kate Beard.

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
