## [Decision Letter · Decision Letter 0]

29 Dec 2020

PONE-D-20-26698

Space-Time Surveillance of COVID-19 Emerging Hotspots using Prospective Scan Statistics Enhanced by Spatiotemporal Event Sequence Based Clustering

PLOS ONE

Dear Dr. Beard,

Thank you for submitting your manuscript to PLOS ONE. After careful consideration, we feel that it has merit but does not fully meet PLOS ONE’s publication criteria as it currently stands. Therefore, we invite you to submit a revised version of the manuscript that addresses all the issues raised by the two reviewers

We look forward to receiving your revised manuscript.

Kind regards,

Agricola Odoi, BVM, MSc, PhD, FAHA, FACE

Academic Editor

PLOS ONE

Journal Requirements:

4. We note that Figures 1- 8 in your submission contain map images which may be copyrighted. All PLOS content is published under the Creative Commons Attribution License (CC BY 4.0), which means that the manuscript, images, and Supporting Information files will be freely available online, and any third party is permitted to access, download, copy, distribute, and use these materials in any way, even commercially, with proper attribution. For these reasons, we cannot publish previously copyrighted maps or satellite images created using proprietary data, such as Google software (Google Maps, Street View, and Earth). For more information, see our copyright guidelines: http://journals.plos.org/plosone/s/licenses-and-copyright.

4.1.    You may seek permission from the original copyright holder of Figures 1-8 to publish the content specifically under the CC BY 4.0 license. 

4.2.    If you are unable to obtain permission from the original copyright holder to publish these figures under the CC BY 4.0 license or if the copyright holder’s requirements are incompatible with the CC BY 4.0 license, please either i) remove the figure or ii) supply a replacement figure that complies with the CC BY 4.0 license. Please check copyright information on all replacement figures and update the figure caption with source information. If applicable, please specify in the figure caption text when a figure is similar but not identical to the original image and is therefore for illustrative purposes only.

Reviewers' comments:

Reviewer's Responses to Questions

**Comments to the Author**

1. Is the manuscript technically sound, and do the data support the conclusions?

Reviewer #1: Partly

Reviewer #2: Partly

2. Has the statistical analysis been performed appropriately and rigorously? 

Reviewer #1: Yes

Reviewer #2: Yes

3. Have the authors made all data underlying the findings in their manuscript fully available?

Reviewer #1: Yes

Reviewer #2: Yes

4. Is the manuscript presented in an intelligible fashion and written in standard English?

Reviewer #1: Yes

Reviewer #2: Yes

5. Review Comments to the Author

Reviewer #1: The authors of the study used prospective space-time scan statistic on number of positive COVID-19 cases in counties to determine location and period of most likely space-time clusters in the continental US. Case counts from counties in most likely clusters were then analyzed through hierarchical agglomerative clustering method to further characterize groups of counties with similar patterns in how frequency of cases (standardized by population size) develop over time. Demographic factors were then used in order to evaluate any differences between clusters. Researches reported that counties detected in the same space-time clusters were classified into different hierarchical clusters. This is interesting finding, and a topic worthy of investigation; although not completely surprising. When methods are combined, this could be done in many different ways and authors used one possible approach - at the very end of the manuscript, they offered some alternatives. This alternative approach was something that I was wondering from the very beginning (no actionable comment here).

In my opinion, by far the largest limitation of the manuscript is its length. Manuscript with 38 pages of text (including tables) is more a technical report than a classical journal article. This is clearly a consequence of thorough analysis, but reading of such contribution is very demanding on the reader, particularly when a lot of technical results consists of reporting counties in most likely clusters. This was done for different time periods, followed by hierarchical clustering. Of course, number of clusters in spatial analysis and in multivariate analysis will be different and this further complicates reading. Is it possible to present an analysis for only one time period (the one authors feel is the most informative), and for the rest of results to be offered in the supplementary material? With a short commentary point on how are other results similar or dissimilar? Also, in multivariate cluster analysis, it is often attempted to explain nature of clusters identified. Although this was attempted to some degree here, this information was diluted among many other details.

Because of manuscript length, it is easy to miss some technical details.

For example.

L104 - are the authors suggesting that RR is available for each location in the cluster? I am not sure that this is in line with how scan statistics work. The RR risk is estimated for the population in the scanning window, or maybe I missed the point here.

L116. Is it possible to determine false positives based on results from scan statistics? Could RR<1 be indication of cluster of cases with lower than expected risk, rather than false positive?

L121. Perhaps check wording. Is it all counties with high RR or all counties from significant clusters with high RR?

L159. Is "severe" the best adjective to use. severe typically refers to clinical expression, whereas cases are just reflection of incidence, some are likely asymptomatic.

L163. Authors should perhaps explain whether they used Poisson model or space-time permutation model. They are different methods.

L163. In this section, it would also be useful to explain how is prospective statistic different from retrospective which is commonly applied in retrospective research studies.

In the methods section - authors did not mention division into different time periods, and what was the rationale for that. reader learns about that in the results section.

L239. "spread" may not be the best word to use here. Clusters do not spread.

Throughout the manuscript, some wording should be improved. e.g. L384 writes about 4 statistically significant counties. is it counties or clusters that are significant?

In addition, the authors try to compare demographics across different clusters, but are not trying to do any statistical testing which could be helpful as a decision point. Is there a reason for that? Are the authors concerned that many comparisons could lead to some false findings?

Reviewer #2: Manuscript #: PONE-D-20-26698

Title: Space-time surveillance of COVID-19 emerging hotspots using prospective scan statistics enhanced by spatiotemporal event sequence based clustering.

Authors: Fuyu Xu & Kate Beard

General comments:

Xu and Beard have used an innovative approach to classifying the temporal patterns of “epidemic curves” at the county level that could make a very important contribution to the epidemiological and disease surveillance literature. Unfortunately, the current manuscript suffers from too much repetition, taking on too many objectives, and perhaps taking a less than ideal approach for integrating their event sequence based clustering with scan statistics. Specifically, the following should be addressed:

1. While there are a number of good reasons to conduct prospective scans, in the context of this manuscript repeating the analysis several times for different periods during the pandemic in the contiguous US states makes the manuscript exceedingly repetitive. I would recommend the authors pick a single period for the analysis and use it consistently throughout the manuscript to exemplify their approach.

2. In terms of the use of the scan statistic, there are a number of major issues that should be addressed by the authors:

i. The decision to limit the maximum size of a cluster to 10% of the population appears to be arbitrary. I would recommend using the default of 50% or less (also the maximum that can be used) of the population. This value does not prevent smaller clusters from being detected, but prevents the need for arbitrary values and is the whole point of the flexible scanning window. In reviewing the figures, it is clear that during certain periods a large number of small clusters are likely part of a larger cluster.

ii. The authors need to clearly state what rule was used for reporting space-time clusters in terms of spatial overlap and preferably spatio-temporal overlap.

iii. The authors are unclear whether they are using a Poisson model or the space-time permutation model for their space-time scans. On lines 166-167, they refer to the permutation model and later to the Poisson distribution on line 185. The space-time permutation model is not based on a Poisson distribution and only requires case data.

iv. If the authors used a Poisson model in SaTScan, it is unclear why they did not adjust for age and sex; this type of standardization is common in most epidemiological analyses. If they used a space-time permutation model, they should recognize that these models identify space-time clusters while inherently adjusting for purely spatial and temporal clusters; in other words this model would adjust for demographic and socio-economic factors if they were related to spatial location.

3. It is unclear why the authors did not apply their event sequence based clustering to all counties. The value seems to be diminished by only applying the technique to locations within active space-time clusters. In fact, investigating if there were spatial clusters of these event sequence clusters, using a multinomial model, would have been a very interesting and perhaps a more appropriate way to combine the two approaches and comment on whether or not these event sequence clusters were randomly distributed or had particular “hotspots”. It would certainly be interesting to compare space-time “hotspots” for rates of disease with spatial clusters of these event sequence clusters.

4. The event sequence clusters shared in geographically distant regions (e.g., Pacific Northwest and New York) likely reflect when COVID-19 was introduced into the US even though molecular sequencing suggests the introduction in these regions was likely from different continents of origin (i.e., Europe vs. Asia). Some discussion concerning the epidemiological interpretation of clusters based on the greater literature is warranted.

5. The socio-demographic analyses should probably be removed from this manuscript. I would recommend performing these analyses using multivariable multinomial regression models based on the event sequence cluster classification for each county in a separate manuscript. Currently, the descriptive comments concerning socio-demographic factors within clusters identified using different methods are not particularly insightful in the current draft of the manuscript. I suspect the authors have put too much in one manuscript and this section deserves much more detail and a stronger analysis.

Specific comments:

Title:

i. The event sequence based clustering was applied to counties during a specific time period, but it did not account for spatial location. It might be better for the authors to state that they are examining the distribution of counties classified based on event sequence based clustering with respect to space-time clusters of COVID-19. If they agree with comment 3 in my general comments, it might be better to focus on the spatial clustering of event sequence based clusters of COVID-19.

Introduction:

i. Lines 60-62: This statement is not really correct. Where and when these measures were implemented, there was success in “flattening the curve” (even in the US), and strict measures did control the disease in parts of the world where the political, economic, and sociological conditions allowed for their strict implementation. I would remove this sentence since it is not relevant to the authors’ work or the need for surveillance tools for the continuing pandemic.

ii. Lines 97-98. This statement is not accurate. The space-time permutation model that is available with SaTScan does not use or require background population data although as a result, it is subject to population shift bias.

iii. Lines 108-109. There’s nothing that prevents an individual from making these comparisons. Please note that the point of the scan statistic is to detect if there are clusters in space, time, and space-time with significantly higher or lower levels of disease without predefining the geographical or temporal size of these clusters. There is no implication that every sub-region within a cluster shares the same rates any more than one could assume that the rate of disease among cities, towns, or villages within a county were homogeneous.

iv. Lines 110-129. This reads more like a summary of methods. I would strongly suggest the authors clearly state their research objectives (i.e., what are they trying to discover or compare rather than a brief description of the methods).

Methods:

i. Please review suggestions in the general comments especially concerning the scan statistic.

ii. Lines 156-161. The authors should use appropriate epidemiological terms. The authors need to clearly state they are calculating the crude incidence rate. However, I would strongly encourage them to consider calculating the age and sex adjusted incidence rates for their subsequent event sequence based clustering.

iii. Lines 183-184. The authors state that the duration of a cluster was set at 2 days, but in the results they have space-time clusters of 1 day in length. The authors should revise the methods or results for consistency.

iv. Line 188. Please note the number of Monte Carlo replications performed and whether scans were performed as 1-tailed tests needs to be stated.

v. Line 198. Please replace the phrase “cases normalized to the county population” with the term incidence rate throughout the manuscript.

Results:

i. Please make certain tables of clusters have consistent titles and select one short form to differentiate event sequence based clusters from space-time clusters and use it consistently throughout the manuscript. Please avoid repeating the methods in the results and use subheadings to differentiate the different statistical approaches being used. Avoid including discussion in the results section.

ii. The tables and figures provide a great deal of summary detail. Please use the text to describe general locations and major characteristics of clusters. The listing of each county within a cluster or each county with a high rate of disease in the text is not necessary. For tables of space-time clusters, please include a column for the radius, the column for the log likelihood in unnecessary. Some authors would include latitude and longitude in these tables, but the figures are sufficient for spatial information.

iii. The figures concerning the “elbow method” and the dendrograms should be moved from the supplemental material into the main manuscript. If the authors follow the suggestions in the general comments, this figure would only be needed for the one period being examined.

iv. Line 339. Please remove the term “emerging”.

v. Line 450. Please note that it is the cluster that is statistically significant and not the counties. Please remove similar phrases from the text.

vi. Line 473. Figures and tables need to be numbered in the order they appear in the text. Currently, figure 9 is listed before figures 7 and 8.

vii. Line 522-523. Please remove results that are not statistically significant from the text and figures.

Discussion:

i. Please revise the discussion after revising the manuscript.

References:

i. Please make certain the references are consistently formatted and all information is included. For instance, the formatting of journal titles in terms of capital letter is inconsistent and the journal is missing from some references.

Tables and figures

i. Please do not include Excel files in the supplemental material. If you believe this material is useful for the reader, generate tables in PDF format with proper variable names and footnotes for any short forms.

6. PLOS authors have the option to publish the peer review history of their article (what does this mean?). If published, this will include your full peer review and any attached files.

Reviewer #1: No

Reviewer #2: No

---

## [Author Response · Author response to Decision Letter 0]

12 Feb 2021

Response to reviewers

Reviewer #1: The authors of the study used prospective space-time scan statistic on number of positive COVID-19 cases in counties to determine location and period of most likely space-time clusters in the continental US. Case counts from counties in most likely clusters were then analyzed through hierarchical agglomerative clustering method to further characterize groups of counties with similar patterns in how frequency of cases (standardized by population size) develop over time. Demographic factors were then used in order to evaluate any differences between clusters. Researches reported that counties detected in the same space-time clusters were classified into different hierarchical clusters. This is interesting finding, and a topic worthy of investigation; although not completely surprising. When methods are combined, this could be done in many different ways and authors used one possible approach - at the very end of the manuscript, they offered some alternatives. This alternative approach was something that I was wondering from the very beginning (no actionable comment here).

In my opinion, by far the largest limitation of the manuscript is its length. Manuscript with 38 pages of text (including tables) is more a technical report than a classical journal article. This is clearly a consequence of thorough analysis, but reading of such contribution is very demanding on the reader, particularly when a lot of technical results consists of reporting counties in most likely clusters. This was done for different time periods, followed by hierarchical clustering. Of course, number of clusters in spatial analysis and in multivariate analysis will be different and this further complicates reading. Is it possible to present an analysis for only one time period (the one authors feel is the most informative), and for the rest of results to be offered in the supplementary material? With a short commentary point on how are other results similar or dissimilar? Also, in multivariate cluster analysis, it is often attempted to explain nature of clusters identified. Although this was attempted to some degree here, this information was diluted among many other details. We recognize that there was too much redundancy and detailing of clusters in the initial manuscript. We did consider a focus on a single study period, but we feel there are some interesting comparisons to convey over the four periods. To be responsive to the reviewer’s concern of length and redundancy we have substantially simplified the results section to focus on a more directed comparison of the two approaches.

Because of manuscript length, it is easy to miss some technical details.

For example.

L104 - are the authors suggesting that RR is available for each location in the cluster? I am not sure that this is in line with how scan statistics work. The RR risk is estimated for the population in the scanning window, or maybe I missed the point here. 

SatScan does compute the RR for individual locations within a cluster as well as the cluster RR. The RR for a county within a cluster is calculated using the following equation as used in Hohl et al 2020

〖RR〗_cty=(c/e)/((C-c)(C-e))where c= total number of cases in a county, C is the total number of observed cases in the conterminous US, e is the expected number of cases in a county calculated as e=p_cty*C/P

L116. Is it possible to determine false positives based on results from scan statistics? Could RR<1 be indication of cluster of cases with lower than expected risk, rather than false positive? 

A RR <1 would not be considered a false positive in the sense of the RR for the cluster. Our thinking here is that in the context of a detected cluster, a specific county within the cluster with a county RR <1 might be construed as a false positive. We have however changed the text to avoid any misrepresentation in this regard.

L121. Perhaps check wording. Is it all counties with high RR or all counties from significant clusters with high RR? Individual counties with RR>1 computed as indicated above were used in the STES analysis.

L159. Is "severe" the best adjective to use. severe typically refers to clinical expression, whereas cases are just reflection of incidence, some are likely asymptomatic. We agree that this term is not appropriate here and have removed this text.

L163. Authors should perhaps explain whether they used Poisson model or space-time permutation model. They are different methods. The prospective Poisson space-time models was used. We apologize for the oversight and ambiguity. 

L163. In this section, it would also be useful to explain how is prospective statistic different from retrospective which is commonly applied in retrospective research studies. A more specific distinction between them has been added to the text.

In the methods section - authors did not mention division into different time periods, and what was the rationale for that. reader learns about that in the results section. In the revised manuscript we have explicitly noted the study time period in the Methods section with some justification for these period intervals.

L239. "spread" may not be the best word to use here. Clusters do not spread. We agree and have changed the text to - as the drivers for these identified clusters.

Throughout the manuscript, some wording should be improved. e.g. L384 writes about 4 statistically significant counties. is it counties or clusters that are significant? We agree and have removed the incorrect reference to statistically significant counties.

In addition, the authors try to compare demographics across different clusters, but are not trying to do any statistical testing which could be helpful as a decision point. Is there a reason for that? 

Are the authors concerned that many comparisons could lead to some false findings? In the revisions to the manuscript, we have removed the demographic analysis section as suggested by reviewer 2. 

Reviewer #2: Manuscript #: PONE-D-20-26698

Title: Space-time surveillance of COVID-19 emerging hotspots using prospective scan statistics enhanced by spatiotemporal event sequence based clustering.

Authors: Fuyu Xu & Kate Beard

General comments:

Xu and Beard have used an innovative approach to classifying the temporal patterns of “epidemic curves” at the county level that could make a very important contribution to the epidemiological and disease surveillance literature. Unfortunately, the current manuscript suffers from too much repetition, taking on too many objectives, and perhaps taking a less than ideal approach for integrating their event sequence based clustering with scan statistics. Specifically, the following should be addressed: 

We recognize that there were too many objectives, too much redundancy and detailing of clusters in the initial manuscript. We did consider a focus on a single study period, but we feel there are some interesting comparisons to convey over the four periods, so we have retained the four study period comparison. To be responsive to the reviewer’s concern of length and redundancy we have substantially simplified the results section to focus on a more directed comparison of the two approaches.

1. While there are a number of good reasons to conduct prospective scans, in the context of this manuscript repeating the analysis several times for different periods during the pandemic in the contiguous US states makes the manuscript exceedingly repetitive. I would recommend the authors pick a single period for the analysis and use it consistently throughout the manuscript to exemplify their approach. We have revised the manuscript to remove the redundancies of describing each period in detail. We feel that the comparison between the space-time scan and the sequence similarity clusters benefits from a comparison over a sequence of time periods. Thus, we have retained the 4 study periods but limit the results and discuss to key comparisons of what is conveyed in the space-time scan view versus the temporal view conveyed by the sequence similarity.

2. In terms of the use of the scan statistic, there are a number of major issues that should be addressed by the authors:

i. The decision to limit the maximum size of a cluster to 10% of the population appears to be arbitrary. I would recommend using the default of 50% or less (also the maximum that can be used) of the population. This value does not prevent smaller clusters from being detected, but prevents the need for arbitrary values and is the whole point of the flexible scanning window. In reviewing the figures, it is clear that during certain periods a large number of small clusters are likely part of a larger cluster. Determining a specific upper bound of scanning window size has been explained in papers by Kulldorf et al. and SaTScan User Guide 9.6. The optimum maximum size of the scanning window should be determined on case by case. 10% of the population was used in similar research (Holz et al 2020) and it seemed is reasonable to replicate that here. We did experiment with using 50% of population for an upper limit which resulted in some extremely large clusters especially at the early stage of pandemic.

ii. The authors need to clearly state what rule was used for reporting space-time clusters in terms of spatial overlap and preferably spatio-temporal overlap. There is no spatial overlap in clusters in the output results, but temporal overlaps do occur. 

iii. The authors are unclear whether they are using a Poisson model or the space-time permutation model for their space-time scans. On lines 166-167, they refer to the permutation model and later to the Poisson distribution on line 185. The space-time permutation model is not based on a Poisson distribution and only requires case data. The prospective Poisson space-time model was used. We apologize for the oversight and ambiguity and have made clarifications in the text.

iv. If the authors used a Poisson model in SaTScan, it is unclear why they did not adjust for age and sex; this type of standardization is common in most epidemiological analyses. If they used a space-time permutation model, they should recognize that these models identify space-time clusters while inherently adjusting for purely spatial and temporal clusters; in other words this model would adjust for demographic and socio-economic factors if they were related to spatial location. We did not have information on age and sex for confirmed cased so were not able to make these adjustments. We also note in the text that while deaths from COVID 19 are several times higher in older age groups, infections can affect all segments of the population.

3. It is unclear why the authors did not apply their event sequence based clustering to all counties. The value seems to be diminished by only applying the technique to locations within active space-time clusters. In fact, investigating if there were spatial clusters of these event sequence clusters, using a multinomial model, would have been a very interesting and perhaps a more appropriate way to combine the two approaches and comment on whether or not these event sequence clusters were randomly distributed or had particular “hotspots”. It would certainly be interesting to compare space-time “hotspots” for rates of disease with spatial clusters of these event sequence clusters. We did apply the sequence similarity-based clustering to all counties, but in this early period of the pandemic many counties had no cases. We have now included these in one “Outsiders” category in the temporal profiles that have been added to sequence similarity cluster maps. As an objective was to compare differences between space-time scan and sequence similarity clusters, we also felt it was useful to focus on the most active case locations. While some of the sequence similarity clusters exhibit some spatial clustering, an aim of the sequence similarity approach was to offer a temporal view on the locations identified by the space-time scan.

4. The event sequence clusters shared in geographically distant regions (e.g., Pacific Northwest and New York) likely reflect when COVID-19 was introduced into the US even though molecular sequencing suggests the introduction in these regions was likely from different continents of origin (i.e., Europe vs. Asia). Some discussion concerning the epidemiological interpretation of clusters based on the greater literature is warranted. We have added references to research on the different continents of origin noting that isolates from China primarily seeded the original COVID-19 outbreak on the West Coast and that European isolates seeded the pandemic in New York (and the US East Coast).

5. The socio-demographic analyses should probably be removed from this manuscript. I would recommend performing these analyses using multivariable multinomial regression models based on the event sequence cluster classification for each county in a separate manuscript. Currently, the descriptive comments concerning socio-demographic factors within clusters identified using different methods are not particularly insightful in the current draft of the manuscript. I suspect the authors have put too much in one manuscript and this section deserves much more detail and a stronger analysis. We agree that this analysis would be better served in another manuscript and have removed it. 

Specific comments:

Title:

i. The event sequence based clustering was applied to counties during a specific time period, but it did not account for spatial location. It might be better for the authors to state that they are examining the distribution of counties classified based on event sequence based clustering with respect to space-time clusters of COVID-19. If they agree with comment 3 in my general comments, it might be better to focus on the spatial clustering of event sequence based clusters of COVID-19. We have revised the title to reflect changes to: “A comparison of prospective space-time scan statistics and event sequence similarity based clustering for COVID 19 surveillance.”

Additionally, we have made revisions in the manuscript to clarify that the sequence similarity clustering was applied in each study period, revised the maps of these clusters and added timelines to show the temporal profiles of these clusters. Some of the sequence similarity clusters do exhibit spatial clustering of some members but the intent was to examine membership with respect to sequence similarity rather than spatial clustering.

Introduction:

i. Lines 60-62: This statement is not really correct. Where and when these measures were implemented, there was success in “flattening the curve” (even in the US), and strict measures did control the disease in parts of the world where the political, economic, and sociological conditions allowed for their strict implementation. I would remove this sentence since it is not relevant to the authors’ work or the need for surveillance tools for the continuing pandemic. This text has been removed in the revised manuscript.

ii. Lines 97-98. This statement is not accurate. The space-time permutation model that is available with SaTScan does not use or require background population data although as a result, it is subject to population shift bias. We did not use the permutation space-time scan statistic. The prospective Poisson space-time models was used. We apologize for the ambiguity in line 168 and have corrected this oversight in the revised manuscript.

iii. Lines 108-109. There’s nothing that prevents an individual from making these comparisons. Please note that the point of the scan statistic is to detect if there are clusters in space, time, and space-time with significantly higher or lower levels of disease without predefining the geographical or temporal size of these clusters. There is no implication that every sub-region within a cluster shares the same rates any more than one could assume that the rate of disease among cities, towns, or villages within a county were homogeneous. The text in this section was removed.

iv. Lines 110-129. This reads more like a summary of methods. I would strongly suggest the authors clearly state their research objectives (i.e., what are they trying to discover or compare rather than a brief description of the methods). We have revised this section to specifically address the paper objectives and have moved the methods related text to the Methods section.

Methods:

i. Please review suggestions in the general comments especially concerning the scan statistic.

We have addressed the suggestions in the general comments on the scan statistic.

ii. Lines 156-161. The authors should use appropriate epidemiological terms. The authors need to clearly state they are calculating the crude incidence rate. However, I would strongly encourage them to consider calculating the age and sex adjusted incidence rates for their subsequent event sequence based clustering. We thank the reviewer for this correction and have revised the manuscript to refer to incidence rate. We did not have information on age and sex for confirmed cased so were not able to make these adjustments. We also note in the text that while deaths from COVID 19 are several times higher in older age groups, infections can affect all segments of the population.

iii. Lines 183-184. The authors state that the duration of a cluster was set at 2 days, but in the results they have space-time clusters of 1 day in length. The authors should revise the methods or results for consistency. The setting with 2 days is correct. A date math error occurred when presenting the cluster duration in the table, which we have corrected in the revised manuscript.

iv. Line 188. Please note the number of Monte Carlo replications performed and whether scans were performed as 1-tailed tests needs to be stated. We chose the Standard Monte Carlo for the p-value (<= 0.05) in the SaTScan setting, and 999 simulations were run.

v. Line 198. Please replace the phrase “cases normalized to the county population” with the term incidence rate throughout the manuscript. This has been corrected through-out the revised manuscript.

Results:

i. Please make certain tables of clusters have consistent titles and select one short form to differentiate event sequence based clusters from space-time clusters and use it consistently throughout the manuscript. Please avoid repeating the methods in the results and use subheadings to differentiate the different statistical approaches being used. Avoid including discussion in the results section. We have revised the test to clearly separate results from discussion.

ii. The tables and figures provide a great deal of summary detail. Please use the text to describe general locations and major characteristics of clusters. The listing of each county within a cluster or each county with a high rate of disease in the text is not necessary. For tables of space-time clusters, please include a column for the radius, the column for the log likelihood in unnecessary. Some authors would include latitude and longitude in these tables, but the figures are sufficient for spatial information. We have included a column for the radius and removed the column of log likelihood (LLR).

iii. The figures concerning the “elbow method” and the dendrograms should be moved from the supplemental material into the main manuscript. If the authors follow the suggestions in the general comments, this figure would only be needed for the one period being examined. In the revised manuscript we have moved the figures of the elbow method graphs and cluster dendrograms to the main body.

iv. Line 339. Please remove the term “emerging”. We have removed this term.

v. Line 450. Please note that it is the cluster that is statistically significant and not the counties. Please remove similar phrases from the text. We have corrected this in the revised manuscript.

vi. Line 473. Figures and tables need to be numbered in the order they appear in the text. Currently, figure 9 is listed before figures 7 and 8. We have revised some figures and taken care that they appear in the correct order and in which they are referenced in the text. 

vii. Line 522-523. Please remove results that are not statistically significant from the text and figures. We have removed the non-statistically significant clusters in the figures and the corresponding description in the text. 

Discussion:

i. Please revise the discussion after revising the manuscript. The discussion has been revised to reflect revisions to the manuscript.

---

## [Decision Letter · Decision Letter 1]

18 Mar 2021

PONE-D-20-26698R1

A comparison of prospective space-time scan statistics and event sequence similarity-based clustering for COVID 19 surveillance

PLOS ONE

Dear Dr. Beard,

Thank you for submitting your manuscript to PLOS ONE. After careful consideration, we feel that it has merit but does not fully meet PLOS ONE’s publication criteria as it currently stands. Therefore, we invite you to submit a revised version of the manuscript that addresses all the issues raised by both reviewers.

We look forward to receiving your revised manuscript.

Kind regards,

Agricola Odoi, BVM, MSc, PhD, FAHA, FACE

Academic Editor

PLOS ONE

Journal Requirements:

Reviewers' comments:

Reviewer's Responses to Questions

**Comments to the Author**

1. If the authors have adequately addressed your comments raised in a previous round of review and you feel that this manuscript is now acceptable for publication, you may indicate that here to bypass the “Comments to the Author” section, enter your conflict of interest statement in the “Confidential to Editor” section, and submit your "Accept" recommendation.

Reviewer #1: All comments have been addressed

Reviewer #2: (No Response)

2. Is the manuscript technically sound, and do the data support the conclusions?

Reviewer #1: Yes

Reviewer #2: Yes

3. Has the statistical analysis been performed appropriately and rigorously? 

Reviewer #1: Yes

Reviewer #2: Yes

4. Have the authors made all data underlying the findings in their manuscript fully available?

Reviewer #1: Yes

Reviewer #2: Yes

5. Is the manuscript presented in an intelligible fashion and written in standard English?

Reviewer #1: Yes

Reviewer #2: Yes

6. Review Comments to the Author

Reviewer #1: Thank you to authors for addressing points raised in the previous review. I only have a couple of minor suggestions for authors to consider.

Authors stated that incubation is 14 days, which is approaching upper limit. Ideally, the authors should carefully word whether this was maximum or average and provide reference.

Next, on images displaying case sequence clusters – it may be useful to indicate, possibly in the figure legend, what are outsiders.

Similarly, for cluster dendrograms it may be helpful to indicate which group of observations belong to which cluster. It seems there is a place for this, but cluster designation is not visible on figures.

In addition, the authors used calendar time to look into similarity of case incidence within clusters. For future consideration, it may actually be interesting to consider time in terms of number of days since detection of the first case in a county.

Reviewer #2: Manuscript ID: PONE-D-20-266698R1

Title: A comparison of prospective space-time scan statistics and event sequence similarity-based clustering of COVID-19 surveillance

Authors: Xu, F. & Beard, K.

General comments: The revised draft of the authors manuscript is greatly improved. Although I am not sure I agree with all their decisions (e.g., maximum scanning window), I believe they have documented/defended their decisions well. My remaining suggestions are mainly cosmetic in nature. Below are some general comments/suggestions:

i. Please put subheadings for the space-time cluster and sequence similarity-based cluster paragraphs in each study period section to avoid confusion over what type of “clusters” are being discussed.

ii. Spell “sequence similarity-based cluster” consistently throughout the manuscript. It is written at least three different ways in the text (e.g., “sequence-similarity based cluster”, “sequence similarity based cluster”).

iii. In paragraphs concerning sequence similarity-based clusters, please do not refer to “spatial clusters” of these clusters. It is very confusing to use the term in a non-statistical sense in a manuscript describing two types of statistical clusters. Just indicate that these sequence similarity-based clusters concentrate around particular cities or regions rather than state they form “spatial clusters”.

iv. In the discussion, make certain to state clearly the value of extracting the information concerning the sequence similarity-based clusters from within the space-time clusters.

Manuscript text:

i. Line 45: It should read “share a similar”.

ii. Line 109: It should read “understanding disease dynamics”.

iii. Lines 113 & 386: It should read “complementary” not “complimentary”.

iv. Line 165: It might be better to write “missing” rather than “avoiding”.

v. Lines 171-173: This statement is not accurate. The age structure of a population will influence disease reporting and the real incidence of disease. The authors should just state they did not have access to age and sex data for cases and were unable to adjust for these variables as they reported in their response letter. This limitation should be addressed in the discussion.

vi. Line 219: Would it be better to state “decreased” rather than “improved”?

vii. Line 263: It should read, “statistical”.

viii. Line 333: Please remove the sentence, “In this period, we note little activity on the west coast.” This statement is not correct. The authors did not identify any active space-time clusters during this period, but there was a lot of disease activity on the West Coast.

ix. Line 347: Replace “covering through” with “ending in”.

x. Line 348: Replace “statistics” with “statistic”.

xi. Line 362: Replace “determined” with “selected”.

xii. Lines 371-372: Make certain to explain what “OC” means in the text.

xiii. Lines 389-390: It might be better to state, “the expected number of cases in space-time based on …...”. The following sentence from 390-391 should be removed since it is redundant with the addition of the above phrase.

xiv. Line 392: It should read, “at such a location during a period of time.”

xv. Line 393: Replace “temporal dimension” with “temporal pattern” since one could argue the “temporal dimension” is part of the space-time cluster.

xvi. Line 400: Rephrase as “we found that in all study periods, similar sequence patterns of COVID-19...”

xvii. Line 408: Consider including “similar changes in surveillance programs” to the list of reasons explaining these common temporal patterns.

xviii. Lines 413-414: Do the authors mean “counties in New York State”? Please clarify.

xix. Line 447: It should read, “pathway studies”.

Tables and figures:

i. Tables 1-4. In the titles, please replace “SaTScan space-time clusters” with “prospective space-time clusters”. A footnote can be added to the tables stating, “Space-time clusters were identified using the spatial scan statistic with a Poisson model”.

ii. Fig 1. Remove “covering” from the title.

iii. I believe the journal expects supplementary materials to be labeled as “Table S1” and “Fig. S1” rather than “S1 Table” or “S1 Figure”. Please correct accordingly.

References:

i. Please properly edit the references. Journal titles and manuscript titles are inconsistently formatted. Additional “PubMed” information is sometimes accidentally included at the end of references. If the authors wish to include “doi” information, please include it consistently or not at all.

ii. Reference 36 should be replaced with a more formal reference (e.g., journal article or government report).

7. PLOS authors have the option to publish the peer review history of their article (what does this mean?). If published, this will include your full peer review and any attached files.

Reviewer #1: No

Reviewer #2: No

---

## [Author Response · Author response to Decision Letter 1]

17 Apr 2021

Reviewer #1: Thank you to authors for addressing points raised in the previous review. I only have a couple of minor suggestions for authors to consider.

Authors stated that incubation is 14 days, which is approaching upper limit. Ideally, the authors should carefully word whether this was maximum or average and provide reference. Thanks for pointing this out. We cited a reference for this in which the incubation time for COVID-19 is mostly ranging from 1-14 days with the average of 5 days.

Next, on images displaying case sequence clusters – it may be useful to indicate, possibly in the figure legend, what are outsiders. We added text in the methods section on comparison to the group outside the clusters which is the Outsiders or OC group. This is also now noted in the captions of all four related maps.

Similarly, for cluster dendrograms it may be helpful to indicate which group of observations belong to which cluster. It seems there is a place for this, but cluster designation is not visible on figures. We added the cluster numbers to the dendrogram figures to facilitate reference between them and the maps. They are also consistently color coded now between figures.

In addition, the authors used calendar time to look into similarity of case incidence within clusters. For future consideration, it may actually be interesting to consider time in terms of number of days since detection of the first case in a county. In this paper we emphasize the purpose of surveillance, so using calendar time seemed appropriate. We agree that using the number of days since detection of the first case in a county for a time measure is an interesting approach for future research. As a matter of fact, we thought this before and believe that using the time counting from the first incidence detection in a county is better for identifying the similar patterns in terms of the development of covid-19 disease itself. 

Reviewer #2: Manuscript ID: PONE-D-20-266698R1

Title: A comparison of prospective space-time scan statistics and event sequence similarity-based clustering of COVID-19 surveillance

Authors: Xu, F. & Beard, K.

General comments: The revised draft of the authors manuscript is greatly improved. Although I am not sure I agree with all their decisions (e.g., maximum scanning window), I believe they have documented/defended their decisions well. My remaining suggestions are mainly cosmetic in nature. Below are some general comments/suggestions:

i. Please put subheadings for the space-time cluster and sequence similarity-based cluster paragraphs in each study period section to avoid confusion over what type of “clusters” are being discussed. We added subheading for space-time scan cluster and sequence similarity-based cluster paragraphs for each of the study periods.

ii. Spell “sequence similarity-based cluster” consistently throughout the manuscript. It is written at least three different ways in the text (e.g., “sequence-similarity based cluster”, “sequence similarity based cluster”). Thank you for catching this. We made the consistent use of “sequence similarity-based cluster” throughout the manuscript.

iii. In paragraphs concerning sequence similarity-based clusters, please do not refer to “spatial clusters” of these clusters. It is very confusing to use the term in a non-statistical sense in a manuscript describing two types of statistical clusters. Just indicate that these sequence similarity-based clusters concentrate around particular cities or regions rather than state they form “spatial clusters”. When some members or counties of sequence similarity-based clusters are spatially clustered we changed the expression to “some members within this group appear spatially concentrated or grouped around metropolitan areas …”.

iv. In the discussion, make certain to state clearly the value of extracting the information concerning the sequence similarity-based clusters from within the space-time clusters.? We have added a paragraph to the methods section on comparison to address this issue. 

Manuscript text:

i. Line 45: It should read “share a similar”. “share similar” has been replaced with “share a similar”.

ii. Line 109: It should read “understanding disease dynamics”. “understanding of the disease dynamics” has been replaced with “understanding disease dynamics”.

iii. Lines 113 & 386: It should read “complementary” not “complimentary”. “complimentary” has been corrected with “complementary”.

iv. Line 165: It might be better to write “missing” rather than “avoiding”. In this case we did want to avoid very large clusters such as ones covering over a quarter of the country as these are not particularly meaningful as “clusters”

v. Lines 171-173: This statement is not accurate. The age structure of a population will influence disease reporting and the real incidence of disease. The authors should just state they did not have access to age and sex data for cases and were unable to adjust for these variables as they reported in their response letter. This limitation should be addressed in the discussion. We agree the effect of age structure of a population on the actual incidence of disease. We added “we were unable to access age and sex data at this time for cases in this study, thus we did not adjust for age and sex”. 

vi. Line 219: Would it be better to state “decreased” rather than “improved”? We made this change

vii. Line 263: It should read, “statistical”. We made this change

viii. Line 333: Please remove the sentence, “In this period, we note little activity on the west coast.” This statement is not correct. The authors did not identify any active space-time clusters during this period, but there was a lot of disease activity on the West Coast. We agree with this and the sentence has been deleted. 

ix. Line 347: Replace “covering through” with “ending in”. We replaced “covering through” with “ending in” since followed by May 20, 2020.

x. Line 348: Replace “statistics” with “statistic”. We replaced “statistics” with “statistic”.

xi. Line 362: Replace “determined” with “selected”. “determined” has been replaced with “selected”. (note: “determined” was in Line 372)

xii. Lines 371-372: Make certain to explain what “OC” means in the text. We added a paragraph in the methods section to explain OC and additional added an explanation of “Outsiders” (OC) in the captions of all four related maps.

xiii. Lines 389-390: It might be better to state, “the expected number of cases in space-time based on …...”. The following sentence from 390-391 should be removed since it is redundant with the addition of the above phrase. We added “in space-time” after “the expected number of cases” and deleted the following sentence.

xiv. Line 392: It should read, “at such a location during a period of time.” We replaced “at such location and times” with “at such a location during a period of time”.

xv. Line 393: Replace “temporal dimension” with “temporal pattern” since one could argue the “temporal dimension” is part of the space-time cluster. Good point, we replaced “temporal dimension” with “temporal pattern”.

xvi. Line 400: Rephrase as “we found that in all study periods, similar sequence patterns of COVID-19...”. We agreed that this is a better expression so we replaced “all study periods showed that similar sequences in COVID-19 cases” with “we found that in all study periods, similar sequence patterns of COVID-19 cases”.

xvii. Line 408: Consider including “similar changes in surveillance programs” to the list of reasons explaining these common temporal patterns. We agreed with the reviewer adding this to the list for one of the reasons.

xviii. Lines 413-414: Do the authors mean “counties in New York State”? Please clarify.There is a New York county so we changed the order of the statement to read Bronx, Kings, Queens, New York, and Wassau counties in New York State, for better clarification.

xix. Line 447: It should read, “pathway studies”. We replaced “pathways studies” with “pathway studies”.

Tables and figures:

i. Tables 1-4. In the titles, please replace “SaTScan space-time clusters” with “prospective space-time clusters”. A footnote can be added to the tables stating, “Space-time clusters were identified using the spatial scan statistic with a Poisson model”. We replaced “SaTScan” with “prospective” and placed the “Note: Space-time clusters were identified using the spatial scan statistic with a Poisson model” below the tables as a footnote.

ii. Fig 1. Remove “covering” from the title. We deleted “covering”.

iii. I believe the journal expects supplementary materials to be labeled as “Table S1” and “Fig. S1” rather than “S1 Table” or “S1 Figure”. Please correct accordingly. We double checked the author guidelines which state the following: “You may use almost any description as the item name of your supporting information as long as it contains an "S" and number. For example, “S1 Appendix” and “S2 Appendix,” “S1 Table” and “S2 Table,” and so forth”. So we followed this guideline. However, the way of labeling that the reviewer suggested may also be accepted.

References:

i. Please properly edit the references. Journal titles and manuscript titles are inconsistently formatted. Additional “PubMed” information is sometimes accidentally included at the end of references. If the authors wish to include “doi” information, please include it consistently or not at all. Manually rechecked and properly edited the Endnote references according to PLoSOne author guidelines. Removed “PubMed” and “doi” information from some references and updated and formatted all references consistently.

ii. Reference 36 should be replaced with a more formal reference (e.g., journal article or government report). The previous Ref 36 was removed and replaced with two recent formal references (Ref 36 and 37).

---

## [Decision Letter · Decision Letter 2]

18 May 2021

PONE-D-20-26698R2

A comparison of prospective space-time scan statistics and spatiotemporal event sequence similarity-based clustering for COVID 19 surveillance

PLOS ONE

Dear Dr. Beard,

Thank you for submitting your manuscript to PLOS ONE. After careful consideration, we feel that it has merit and will most likely be accepted after the minor revisions suggested by reviewer 2 have been implemented. Therefore, we invite you to submit a revised version of the manuscript after making the recommended revisions.

Please submit your revised manuscript by Jul 02 2021 11:59PM.  Please include the following items when submitting your revised manuscript:

We look forward to receiving your revised manuscript.

Kind regards,

Agricola Odoi, BVM, MSc, PhD, FAHA, FACE

Academic Editor

PLOS ONE

Journal Requirements:

Reviewers' comments:

Reviewer's Responses to Questions

**Comments to the Author**

1. If the authors have adequately addressed your comments raised in a previous round of review and you feel that this manuscript is now acceptable for publication, you may indicate that here to bypass the “Comments to the Author” section, enter your conflict of interest statement in the “Confidential to Editor” section, and submit your "Accept" recommendation.

Reviewer #2: (No Response)

2. Is the manuscript technically sound, and do the data support the conclusions?

Reviewer #2: Yes

3. Has the statistical analysis been performed appropriately and rigorously? 

Reviewer #2: Yes

4. Have the authors made all data underlying the findings in their manuscript fully available?

Reviewer #2: Yes

5. Is the manuscript presented in an intelligible fashion and written in standard English?

Reviewer #2: Yes

6. Review Comments to the Author

Reviewer #2: Manuscript ID: PONE-D-20-26698R2

Manuscript title: A comparison of prospective space-time scan-statistics and spatiotemporal event sequence-based clustering for COVID 19 surveillance

Corresponding author: Beard-K

General comments:

The authors have made all the requested revisions. Below are some minor edits/suggestions they should consider. There is no need for me to see the manuscript again.

Specific comments:

Line 172: Should it read, “required”?

Line 174-175: It might sound better to write, “....for cases in this study, so we could not adjust for age and sex.”

References:

Fix the formatting of journal titles for references 17, 21, 34 (also the article title).

7. PLOS authors have the option to publish the peer review history of their article (what does this mean?). If published, this will include your full peer review and any attached files.

Reviewer #2: No

---

## [Author Response · Author response to Decision Letter 2]

20 May 2021

We responded to each of the reviews suggestions and made the corresponding changes

---

## [Editor Report · Decision Letter 3]

27 May 2021

A comparison of prospective space-time scan statistics and spatiotemporal event sequence similarity-based clustering for COVID 19 surveillance

PONE-D-20-26698R3

Dear Dr. Beard,

We’re pleased to inform you that your manuscript has been judged scientifically suitable for publication and will be formally accepted for publication once it meets all outstanding technical requirements.

Kind regards,

Agricola Odoi, BVM, MSc, PhD, FAHA, FACE

Academic Editor

PLOS ONE
---

## [Editor Report · Acceptance letter]

2 Jun 2021

PONE-D-20-26698R3 

A comparison of Prospective Space-time Scan Statistics and Spatiotemporal Event Sequence Based Clustering for COVID-19 Surveillance 

Dear Dr. Beard:

I'm pleased to inform you that your manuscript has been deemed suitable for publication in PLOS ONE. Congratulations! Your manuscript is now with our production department. 

Kind regards, 

on behalf of

Prof. Agricola Odoi 

Academic Editor

PLOS ONE